# Somatic NAP1L1 p.D349E promotes cardiac hypertrophy through cGAS-STING-IFN signaling

Cheng Lv[1,5], Xiayidan Alimu[1,5], Xiao Xiao[1,5], Fei Wang[1,5], Jizheng Wang ![ORCID][1,5], Shuiyun Wang[2], Guixin Wu[1], Yu Zhang[1], Yue Wu ![ORCID][3], Houzao Chen ![ORCID][4], Rutai Hui[1], Lei Song ![ORCID][1] ✉ & Yibo Wang ![ORCID][1] ✉

Hypertrophic cardiomyopathy (HCM) is the most common inherited heart disease, often caused by sarcomere gene mutations, though many sporadic cases remain genetically unexplained. Here we show that the somatic variant NAP1L1 p.D349E was involved in cardiac hypertrophy in sporadic HCM patients. Through next generation sequencing, we found that somatic variant NAP1L1 p.D349E was recurrent in the cardiomyocytes of gene-elusive sporadic HCM patients. Subsequent in vivo and in vitro functional analysis confirmed that NAP1L1 p.D349E contributes to HCM by triggering an innate immunity response. This mutation destabilizes nucleosome formation, causing DNA to leak into the cytoplasm. This leakage activates a key immune pathway, cGAS-STING, which leads to the release of inflammatory molecules and promotes heart muscle thickening. Our findings reveal a new mechanism driving HCM and suggest that somatic variants could be important in understanding and management of HCM.

Hypertrophic cardiomyopathy (HCM) is defined as maximum end-diastolic wall thickness ≥15 mm anywhere in the left ventricle on 2-dimensional echocardiography or cardiovascular magnetic resonance (CMR) in the absence of other causes of cardiac hypertrophy[1]. HCM is the most common inherited heart disease, with approximately 36%–64% of cases caused by pathogenic sarcomere variants[2,3]. Patients with a positive family history are more likely to be identified with pathogenic variants than those with a negative family history[4,5]. Lipid metabolism, immunity, and oxidative stress play important roles in HCM[6]. Nevertheless, the genetic etiology of many HCM patients, especially sporadic cases, remains unsolved.

Postzygotic variation occurs continuously during individual development and affects the development of organs and diseases[7]. The crucial role of somatic variants in "benign" diseases, including

some cardiovascular diseases, has been discussed[8]. Somatic variants in single cardiomyocytes were revealed to be related to age-associated DNA damage and oxidative genotoxicity[9], which challenged the theory that somatic variants seldom occur in cardiomyocytes due to the laziness of cell division. However, little attention has been given to somatic variants in patients with HCM.

The cGAS-STING system recognizes conserved molecular patterns and activates downstream immune responses. DNA sensors play important roles in pattern recognition[10,11]. In addition to invading DNA, cGAS-STING can also be activated by mitochondrial DNA (mtDNA) or nuclear DNA damage and leakage[12,13]. The cGAS-STING signaling pathway consists of cyclic GMP-AMP synthetase (cGAS) and stimulator of interferon genes (STING), which is one of the major evolutionarily conserved defense mechanisms that can be used to detect cytoplasmic

[1]State Key Laboratory of Cardiovascular Disease, Fuwai Hospital, National Center for Cardiovascular Diseases, Chinese Academy of Medical Sciences and Peking Union Medical College, Beijing, China. [2]Department of Cardiac Surgery, Fuwai Hospital, Chinese Academy of Medical Sciences and Peking Union Medical College, Beijing, China. [3]Department of Cardiovascular Medicine, The First Affiliated Hospital of Xi'an Jiaotong University, Xi'an, China. [4]State Key Laboratory of Common Mechanism Research for Major Diseases, Department of Biochemistry and Molecular Biology, Institute of Basic Medical Sciences, Chinese Academy of Medical Sciences and Peking Union Medical College, Beijing, China. [5]These authors contributed equally: Cheng Lv, Xiayidan Alimu, Xiao Xiao, Fei Wang, Jizheng Wang. ✉e-mail: songlqd@126.com; yibowang@hotmail.com

double-stranded DNA (dsDNA)[14–16]. cGAS acts as a cytoplasmic dsDNA sensor and generates cGAMP (cyclic GMP-AMP), a cyclic dinucleotide second messenger that activates the adapter protein STING (stimulator of interferon genes) located in the endoplasmic reticulum, which in turn initiates the inflammatory response. Several studies have shown that cGAS-STING signaling plays an essential role in dilated cardiomyopathy, overload-induced heart failure, doxorubicin-induced cardiotoxicity (DIC), and diabetic cardiomyopathy[17–22].

*NAP1L1* (nucleosome assembly protein-like (1) is a member of the NAP1L family and is involved in nucleosome assembly, exchange of histone H2A-H2B dimers, transcriptional regulation, and cell proliferation[23–25]. The role of *NAP1L1* in HCM has rarely been discussed. NAP1-like is involved in efficient transcription-coupled DNA repair[26,27]. Histone density is maintained during transcription by remodeling the structure of chromatin and NAP1 in vitro[28]. We speculated that NAP1L1 dysfunction may interfere with the maintenance of DNA integrity during transcription, which may leak DNA to the cytoplasm and trigger cGAS-STING signaling.

In recent studies, non-classical genetic alterations have been linked to various cardiovascular diseases[8,29]. Through somatic mutation screening in two sporadic HCM cohorts, we addressed the NAP1L1 p.D349E mutation to be a potential contributor to HCM. We propose that the p.D349E mutation destabilizes nucleosome formation, leading to aberrant release of DNA into the cytosol. This release triggers the cGAS-STING pathway, which results in the production of pro-inflammatory cytokines, leading to cardiac hypertrophy by creating a hyper-immune microenvironment. By exploring this mechanism, we aim to uncover new insights into the molecular basis of HCM and identify potential therapeutic targets.

**Table 1 | Comparison of clinical manifestations in gene-elusive sporadic patients and those with P/LP germline variants**

| | Gene-elusive sporadic patients | Those with P/LP germline variants | *p*-value |
|---|---|---|---|
| Number of Patients, *n* | 20 | 29 | |
| Male, *n* | 12 | 16 | 0.97 |
| Age at diagnosis (years) | 45 ± 12.8 | 35.8 ± 15.4 | 0.03 |
| Duration of disease (years) | 7.4 ± 4.7 | 6.9 ± 3.8 | 0.34 |
| BMI (kg/m²) | 19.7 ± 3.5 | 20.9 ± 2.7 | 0.67 |
| Heart Rate, *n* | 77.4 ± 9.4 | 80.4 ± 4.7 | 0.59 |
| Co-morbidities | | | |
| Diabetes, *n* | 4 | 8 | 0.79 |
| Hypercholesterolemia, *n* | 12 | 15 | 0.78 |
| Atrial Fibrillation, *n* | 0 | 3 | 0.38 |
| NSVT, *n* | 2 | 6 | 0.44 |
| Chest Pain, *n* | 6 | 11 | 0.79 |
| Syncope, *n* | 1 | 8 | 0.06 |
| NYHA class III or IV, *n* | 9 | 20 | 0.17 |
| Medications | | | |
| Beta blockers, *n* | 13 | 19 | 0.34 |
| Calcium channel blockers, *n* | 10 | 16 | 0.89 |
| Thickness of IVS (mm) | 18.8 ± 5.3 | 21.1 ± 6.1 | 0.17 |
| Maximal LV wall thickness (mm)ᵃ | 23.4 ± 6.0 | 25.7 ± 6.0 | 0.19 |
| LVEF (%) | 73.5 ± 6.8 | 71.1 ± 8.1 | 0.29 |
| LVEDD (mm) | 44.1 ± 6.0 | 39.9 ± 6.4 | 0.02 |

*IVS* interventricular septum, *LVEF* left ventricular ejection fraction, *LV* left ventricular, *LVEDD* left ventricular end-diastolic dimension, *NSVT* nonsustained ventricular tachycardia, *NYHA* New York Heart Association.
ᵃCombined with cardiac MRI.

## Results

### Seventy-one sporadic HCM patients were recruited as the discovery cohort

To investigate the genetic structure of sporadic HCM, we performed WES and ddPCR in two independent HCM cohorts. For the discovery cohort, we recruited seventy-one sporadic patients with HCM at Fuwai Hospital from 2012 to 2015. Patients undergoing myomectomy were selected based on the presence of symptomatic hypertrophic obstructive cardiomyopathy with significant left ventricular outflow tract (LVOT) obstruction (typically defined as a resting gradient >50 mmHg or provocable gradient >70 mmHg) despite optimal medical therapy. The average resting LVOT gradient for the cohort was 54 ± 23 mmHg. Written informed consent was obtained from each patient or his/her proxy. After 100x WES of peripheral blood and classification of germline variants, we found that twenty-nine (40.85%) patients carried thirty-two P/LP variants (Supplementary Data 1) and that twenty-two (30.99%) carried 29 VUSs in sarcomere-encoding genes (Supplementary Data 2). After excluding these patients, twenty genetically unexplained sporadic HCM patients were ultimately included in the somatic variant calling cohort. The workflow is shown in Supplementary Fig. S1.

Compared to patients with pathogenic HCM-associated P/LP germline variants, sporadic genetically unexplained patients were diagnosed at apparently older ages (mean age 45 ± 12.80 years, range of age at HCM diagnosis 12-63 years; Table 1) and mainly at late onset, 50% of whom were diagnosed or had outflow tract obstruction beyond 50 years old. However, there was no significant difference in the time from diagnosis to the need for surgery for outflow obstruction. In terms of clinical manifestations, the left ventricular end-diastolic dimension was much lower in patients with germline variants ($p < 0.05$). Other parameters, such as the occurrence of syncope, thickness of the IVS, and maximal LV wall thickness, only showed a tendency but with no statistical significance. In conclusion, sporadic genetically unexplained patients were diagnosed at older ages and had a less severe clinical course.

### Forty-nine sporadic genetically unexplained HCM patients were recruited as the replication cohort

Forty-nine sporadic genetically unexplained HCM patients were recruited from the First Affiliated Hospital of Xi'an Jiaotong University as an independent replication cohort. WES (100×) of peripheral blood was performed during previous studies. We used the same conditions to select sporadic genetically unexplained HCM patients. The baseline information of the replication cohort is shown in Supplementary Table S1.

### Recurrent somatic NAP1L1 p.D349E was observed in both cohorts

We focused on recurrent variants that occurred in more than one person. No novel recurrent deleterious germline variants were found in either cohort, so we turned our attention to somatic variants. The somatic variant calling procedure is described in the Methods section. We found two patients carrying the somatic variant *NAP1L1* (NM_004537:c. T1047A:D349E; Fig. 1A, D47, and D6). As a nucleosome assembly protein, NAP1L1 is critical for nucleosome assembly[30]. We speculated that disruption of NAP1L1 function might cause failure of histone deaggregation and reassembly, resulting in genomic instability, which leads to DNA leakage into the cytoplasm and activation of cGAS-STING-IFN signaling. The p.D349E mutation in NAP1L1 was predicted to be damaged by Polyphen2. The CADD score of NAP1L1 p.D349E was 23.1. According to BLAST analysis, the gene homology of NAP1L1 between humans and mice is 92%, and the protein homology is 98%. Among the one hundred vertebrate species, the 1047ᵗʰ base of *NAP1L1* is T or C, and both encode aspartic acid due to degeneracy (Supplementary Data 3). Based on the WES results, we performed

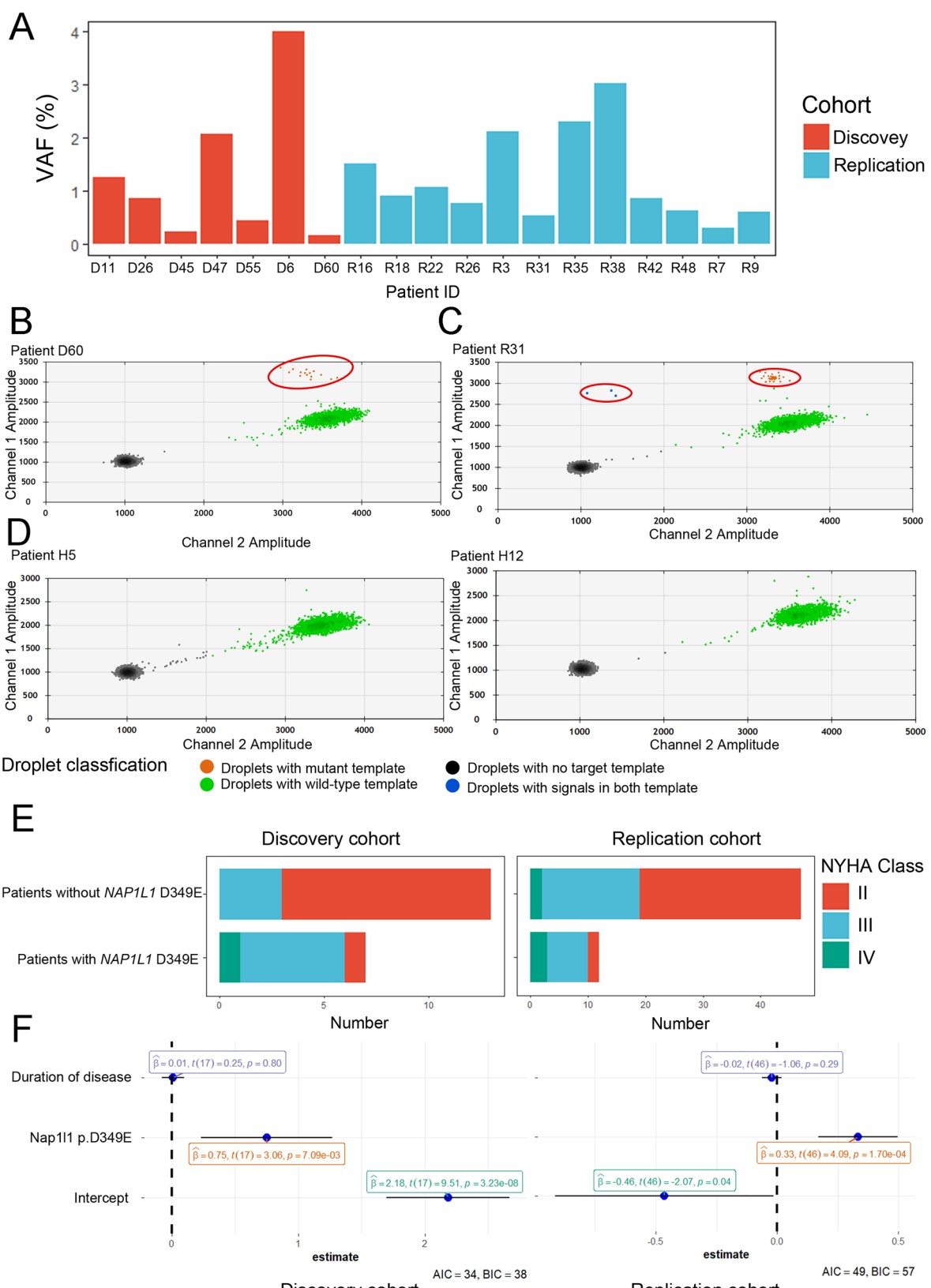

**Droplet classification**
- Droplets with mutant template (orange)
- Droplets with wild-type template (green)
- Droplets with no target template (black)
- Droplets with signals in both template (blue)

droplet digital polymerase chain reaction (ddPCR) to detect somatic variants with lower VAFs in all 20 patients. The results revealed five additional patients harboring somatic variants of NAP1L1 p.D349E (Fig. 1A, B, as a representative).

We then evaluated the subgroup of patients carrying the p.D349E mutation and found that their LVOT gradients were

$57 \pm 21$ mmHg, slightly higher than in patients without any mutation, but lower than in patients with a known germline pathogenic mutation.

In the replication cohort, DNA from cardiac tissues collected during cardiac surgery was subjected to ddPCR with NAP1L1 p.D349E. The ddPCR results showed that twelve of the forty-nine (20%) HCM

**Fig. 1 | Characterization of the Somatic NAP1L1 p.D349E Variant and Its Clinical Associations. A** Variant allele frequency distribution of NAP1L1 p.D349E. Variant allele frequency distribution of *NAP1L1* p.D349E. **B** Representative somatic variants identified by ddPCR in the discovery cohort (D60). X axis, fluorescence intensity detected in the HEX-channel (channel 2, wild type); Y axis, fluorescence intensity detected in the FAM-channel (channel 1, mutant type); gray dots, droplets with background fluorescence; green dots, droplets with fluorescence detected in the HEX-channel; orange dots, droplets with fluorescence detected in the FAM-

channel; blue dots, droplets with signals in both channels. **C** Representative somatic mutations identified by ddPCR in the replication cohort (R31). **D** Representative plot of ddPCR-negative NAP1L1 p.D349E (H5 and H12). **E** Bar plot of the NYHA class in patients with and without somatic *NAP1L1* p.D349E. Left, discovery cohort; **F** Dot-and-whisker plots show the association of the NAP1L1 p.D349E variant with NYHA class in gene-elusive sporadic patients, analyzed using linear regression ($N = 20$ for the discovery cohort, $N = 59$ for the replication cohort). Data are presented as mean values +/− SEM.

patients carried somatic variants of NAP1L1 p.D349E (Fig. 1A, C, as a representative).

To rule out false positives, we tested the ddPCR system in negative controls and no-template controls in different ways, and the probe always presented a clean background. Representative plots of negative controls are shown in Fig. 1D and Supplementary Fig. S2A. Our criterion for calling a ddPCR result positive was that a sample must present at least three positive droplets and must present a number of positive droplets that is at least three times the average number of positive droplets in negative controls to ensure statistical significance.

NAP1L1 p.D349E was in a repeat area, resulting in a potential of ddPCR errors during PCR amplification. The length of the WES reads was approximately 150 bp, enough to overlay the repeat area, so the accuracy of WES was not affected by the repeat area. To validate the ddPCR results, we chose 10 patients with a higher VAF who were positive according to ddPCR; 300x WES was performed on these patients, and 6 patients had mutant p.D349E reads. Considering that the depth of WES is much lower than that of ddPCR, 4 somatic NAP1L1 p.D349E strains with low VAFs were not detected by WES, which is acceptable and sufficient to prove the accuracy of our ddPCR.

Moreover, we applied amplicon sequencing to validate the results of WES and ddPCR. Amplicon sequencing was performed on two samples that were positive by WES and two samples that were positive by ddPCR, and all of the four samples showed true positives (Supplementary Fig. S2B).

We performed a series of ddPCR with low-frequency somatic variants to confirm the sensitivity of the system. The ddPCR assay proved robust quantitative performance across a wide range of variant allele frequencies (VAFs) from 0.03% to 10% (Supplementary Fig. S2C). Under our defined threshold of at least three positive droplets, the assay reliably detected variants at a minimum VAF of 0.03%. The correlation between the expected VAFs and the ddPCR-measured VAFs was strong, with an $R^2$ value of 0.99998, indicating the accuracy of ddPCR in quantifying somatic variants within this range (Supplementary Fig. S2C and Supplementary Table S2).

In all, 35% (7 of 20) of patients in the discovery cohort and 20% (12 of 59) in the replication cohort carried the somatic variant NAP1L1 p.D349E (details in Supplementary Table S3).

## NAP1L1 p.D349E was absent in controls and common human genetic variation databases

None of the patients in either cohort carried the germline variant NAP1L1 p.D349E. We reviewed the variant with IGV and confirmed that all variants had good sequencing quality (Supplementary Fig. S3). Moreover, WES of peripheral blood samples revealed no somatic NAP1L1 p.D349E. We obtained septal biopsies from 27 individuals without HCM as normal controls and performed ddPCR of NAP1L1 p.D349E. The result was negative. NAP1L1 p.D349E was absent in the Genome Aggregation Database, Exome Aggregation Consortium, 1000 Genomes, or NHLBI Exome Sequencing Project. For the somatic variant database, we referred to SomaMutDB, a somatic mutation database of normal human tissues, and no NAP1L1 p.D349E was found in any healthy tissues[31].

## Patients with NAP1L1 p.D349E showed a higher NYHA class

We investigated the correlation between the New York Heart Association (NYHA) Functional Classification and NAP1L1 p.D349E. Of the

seven patients with NAP1L1 p.D349E in the discovery cohort, one was in NYHA class II, five were in NYHA class III, and one was in NYHA class IV. Ten in NYHA class II and three in class III among the thirteen patients did not carry NAP1L1 p.D349E or causative germline variants (Fig. 1E, left). In the replication cohort, two patients in class II, 7 in class III, and three in class IV had NAP1L1 p.D349E, and twenty-eight in class II, seventeen in class III and two in class IV had no NAP1L1 p.D349E (Fig. 1E, right). Due to the limited number of patients, we combined the two cohorts and used the Kendall rank correlation coefficient to assess whether the NYHA class was associated with NAP1L1 p.D349E. The results showed that patients with NAP1L1 p.D349E were in a higher NYHA class ($p = 0.0001179$, tau $= 0.4233794$). We tested other risk factors and found no significant differences. Moreover, the NT-pro BNP level was not significantly different between the two groups. In fact, the result of the Kendall rank correlation coefficient in the separate discovery and replication cohorts also strongly suggested that patients with NAP1L1 p.D349E were in a higher NYHA class ($p = 0.006796$, tau $= 0.6080491$ in the discovery cohort and $p = 0.003331$, tau $= 0.373$ in the replication cohort).

As the disease course prolongs, patients may tend to have a higher NYHA class[32]. We investigated the duration of cardiac disease in these patients, and there was no significant difference between patients with or without NAP1L1 p.D349E. In addition, we conducted a linear model to explore the relationship between the NYHA class variable and the predictors NAP1L1 p.D349E and duration of disease. In the subgroup of gene-elusive sporadic patients in the discovery cohort, obtaining NAP1L1 p.D349E was associated with an increase in the NYHA class ($p = 7.09e\text{-}03$, $\beta_1 = 0.75$; Fig. 1F, left). This correlation was also detected in the replication cohort ($p = 1.07e\text{-}04$, $\beta_1 = 0.33$; Fig. 1F, right).

## NAP1L1 p.D349E mainly occurred in cardiomyocytes

To identify the type of cells in which NAP1L1 p.D349E occurred, we used FACS to sort cTnT+ cells (cardiomyocytes) and cTnT− cells from three cardiac samples from patients D6, R35, and R38 with the highest VAFs. ddPCR analysis showed that the cTnT+ cells had a higher fractional abundance of the variant NAP1L1 p.D349E, which indicated that the somatic variant NAP1L1 p.D349E mainly occurred in cardiomyocytes (Supplementary Fig. S4).

## Knockdown of *NAP1L1* triggered the cytosolic DNA-sensing pathway and interferon activation

To preliminarily investigate the function of *NAP1L1* in cardiomyocytes, we transfected *Nap1l1* siRNA and control siRNA into neonatal rat cardiomyocytes (NRCMs) and performed RNA-Seq. DNA damage response-related genes were upregulated in the *Nap1l1* knockdown group (Supplementary Fig. S5A). Interferon-stimulated genes (ISGs) were upregulated in the *Nap1l1* knockdown group (Supplementary Fig. S5A, B). Cgas was upregulated in the *Nap1l1* knockdown group (Supplementary Fig. S5B). GSEA also revealed that the cytosolic DNA-sensing pathway was upregulated in the *Nap1l1*-knockdown group (Supplementary Fig. S5C). IFN-related pathways, including those related to the response to interferon-alpha, response to interferon-beta, and cellular response to interferon-alpha, were also upregulated in the *Nap1l1* knockdown group (Supplementary Fig. S5D). These results suggested that *Nap1l1* maintains genomic stability and that blocking *Nap1l1* activates cGas-Sting signaling.

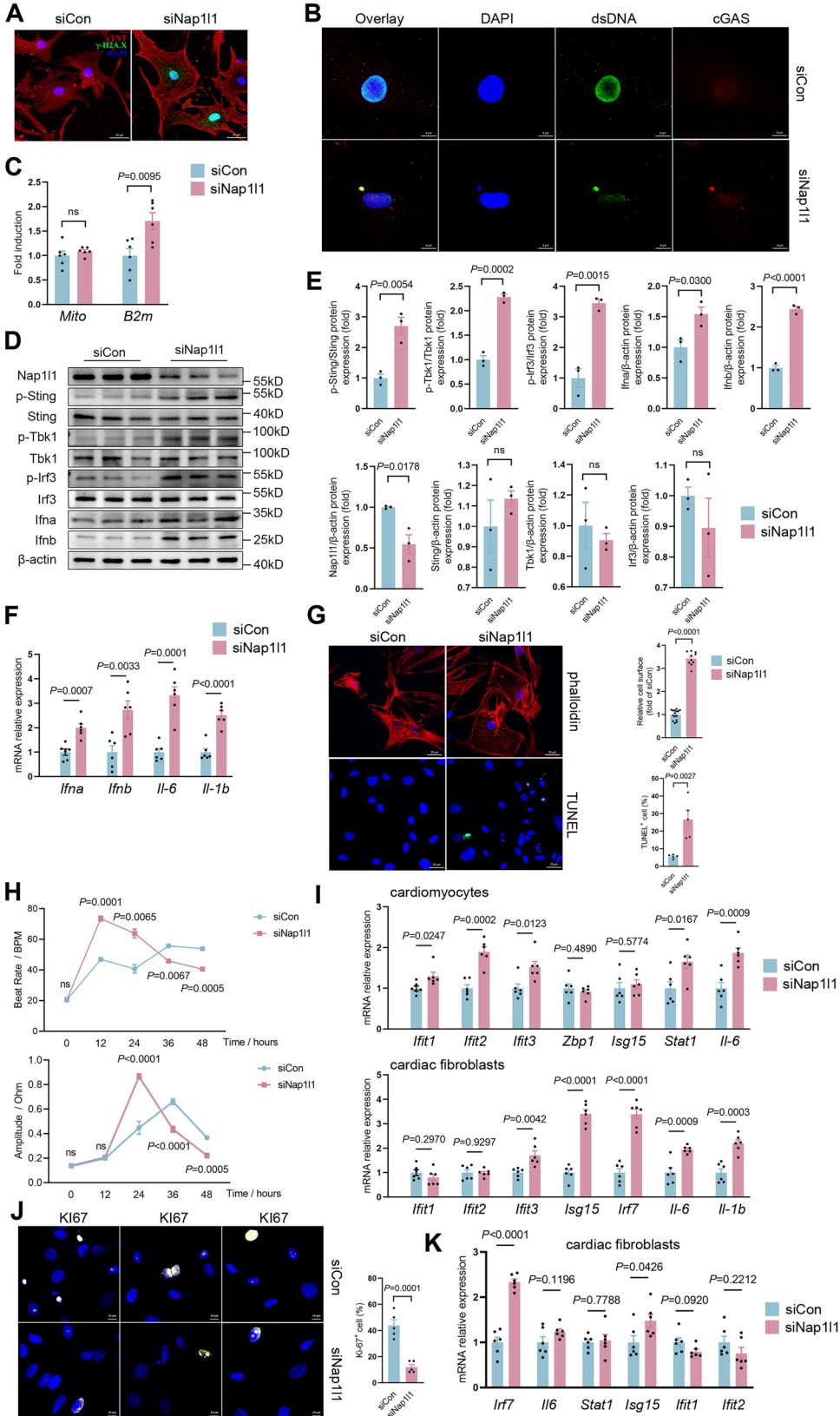

We performed further experiments to validate the results of the RNA-Seq. Immunofluorescence staining revealed that the knockdown of *Nap1l1* increased the level of γ-H2AX in the nucleus (Fig. 2A and Supplementary Fig. S5E). Knockdown of *Nap1l1* resulted in cytoplasmic dsDNA foci and cytoplasmic cGAS localization (Fig. 2B and Supplementary Fig. S5F). To exclude interference from mitochondrial DNA,

we performed whole-cell cytosolic DNA analysis, and the results showed that the increase in cytoplasmic DNA in the si*Nap1l1* group was caused by leakage of nuclear DNA (Fig. 2C). Phosphorylation of Sting, Tbk1, and Irf3 increased in response to knockdown of *Nap1l1* (Fig. 2D, E). As a result, *Ifna, Ifnb, Il-6,* and *Il-1b* were significantly upregulated in the si*Nap1l1* group (Fig. 2F and Supplementary

**Fig. 2 | Effects of Nap1l1 Knockdown on DNA Damage, Inflammatory Signaling, and Cellular Function in NRCMs and Cardiac Fibroblasts.**
**A** Immunofluorescence staining showing the levels of γ-H2ax protein in NRCMs with siCon or siNap1l1; **B** Representative immunofluorescence staining showing the levels of DAPI, dsDNA and cGAS in NRCMs with siCon or siNap1l1; **C** qPCR analysis of Mito and B2m in NRCMs with siCon or siNap1l1. $N = 6$ for each group, and two-tailed unpaired Student's $t$-test was applied; **D, E** Western blot and quantification showing the levels of Sting, Tbk1, Irf3, Ifna, Ifnb, and phosphorylation of Sting, Tbk1, and Irf3 in NRCMs with siCon or siNap1l1. $N = 3$ for each group, and two-tailed unpaired Student's $t$-test was applied; **F** qPCR analysis of Ifna, Ifnb, Il-6, and Il-1b in NRCMs with siCon or siNap1l1. $N = 6$ for each group, and two-tailed unpaired Student's $t$-test was applied; **G** Phalloidin staining and TUNEL (terminal deoxynucleotidyl transferase-mediated dUTP-biotin nick end labeling) staining and quantification in NRCMs with siCon or siNap1l1. Relative cell surface: siCON, $n = 15$; siNap1l1, $n = 10$. TUNEL+ cell: $N = 5$ for each group, and two-tailed unpaired Student's $t$-test was applied; **H** Measurement of heart rate and diastolic function over time after adding siRNA. $N = 4$ for each group and two-way ANOVA with Tukey tests were used for correction of multiple comparisons; **I** qPCR analysis of Ifit1, Ifit2, Ifi3, Zbp1, Isg15, Stat1, and Il6 in influenced cardiomyocytes and Ifit1, Ifit2, Ifi3, Isg15, Irf7, Il6 and Il1b in influenced cardiac fibroblasts. $N = 6$ for each group, and two-tailed unpaired Student's $t$-test was applied; **J** Representative plot of KI67 staining and quantification. $N = 5$ for each group, and two-tailed unpaired Student's $t$-test was applied; **K** qPCR analysis of Irf7, Il6, Stat1, Isg15, Ifit1, and Ifit2 in influenced cardiac fibroblasts. $N = 6$ for each group and a two-tailed unpaired Student's $t$-test was applied. Data are presented as mean values +/− SEM.

Fig. S5G). We performed an apoptosis detection experiment, and the results showed that knocking down *Nap1l1* induced apoptosis (Fig. 2G). To evaluate the functional status of the NRCMs, we conducted heart rate tests and diastolic function tests. The results showed that the heart rate and diastolic function of the *Nap1l1* knockdown group showed an increase in stress, followed by a decrease compared with the control siRNA group (Fig. 2H).

To study paracrine/autocrine effects, we cultured wild-type cardiomyocytes and cardiac fibroblasts in culture medium after culturing NRCMs with *Nap1l1* knockdown or control siRNA for 48 h to simulate the in vivo immune environment (Supplementary Fig. S5H). RT-qPCR results demonstrate that both cardiomyocytes and cardiac fibroblasts can respond to IFN and upregulate the expression of ISGs but with different ISG types (Fig. 2I). *Ifit1, Ifit2, Ifit3, Stat2,* and *Il6* were upregulated in cardiomyocytes, and *Ifit3, Irf7, Il6 and Il-1b* were upregulated in cardiac fibroblasts (Fig. 2I). By immunofluorescence of KI67, we found that *Nap1l1* knockdown resulted in decreased cardiomyocyte proliferation, which is consistent with the impact of *Nap1l1* knockdown in tumors (Fig. 2J)[33,34]. To verify the simulation of paracrine/autocrine, we performed a trans-well assay with membrane separation of control and si*Nap1l1* transfected NRCMs. The results supported the hypothesis of a paracrine effect on WT cardiomyocytes induced by a small number of NAP1l1 p.D349E variant-containing CMs (Fig. 2K and Supplementary Fig. S5I).

## Nap1l1 p.D349E in vivo promoted cardiac hypertrophy by cGas-Sting signaling

To investigate whether NAP1L1 p.D349E mediates cardiac hypertrophy in vivo, we generated recombinant adeno-associated virus serotype-9 (AAV9) harboring the human NAP1L1 p.D349E gene with the cardiac troponin T (cTNT) promoter (AAV9-cTNT- h*NAP1L1* p.D349E-3flag-EGFP) and wild type (AAV9-cTNT- h*NAP1L1*-3flag-EGFP. Fig. 3A). The AAV approach only interferes with a subset of cardiomyocytes, allowing us to simulate the somatic variant with a low VAF. We tested the system, and the virus was expressed only in cTnT+ cardiomyocytes (Supplementary Fig. S6A, B). Mice were treated with Ang II (1 mg/kg/day dissolved in 0.9% NaCl) once daily by subcutaneous injection for 4 weeks beginning 2 days after AAV injection. The NAP1L1 p.D349E AAV-treated mice exhibited significant enlargement in heart weight, heart size, and cardiomyocyte size (Fig. 3B–E). In addition to increased size, cardiomyocytes from NAP1L1 p.D349E AAV-treated mice showed more severe myocardial fibrosis (Fig. 3F, G and Supplementary Fig. S6C). Density of total and small arteriolar was downregulated in MT group (Supplementary Fig. S6D, E). Cardiomyocyte apoptosis was upregulated in the MT group, and macrophage infiltration was increased in the MT group (Supplementary Fig. S6F, G).

For cardiac function assessment, before Ang II injection, there was no difference in left ventricular end-diastolic anterior wall thickness (LVAW;d), left ventricular end-diastolic posterior wall thickness (LVPW;d), ejection fraction (EF), fractional shortening (FS) or left ventricular (LV) mass (Fig. 3H, I). However, after Ang II treatment, the

NAP1L1 p.D349E AAV-treated mice exhibited worse cardiac function, including LVPW;d, LVPW;s, EF, FS, and increased LV mass (Fig. 3I). Common HCM markers, including Anp, Bnp, and β-Mhc, were upregulated in NAP1L1 p.D349E AAV-treated mice with Ang II compared with wild type (Fig. 3J).

We further investigated whether cGAS-Sting signaling was involved in Nap1l1 p.D349E-mediated cardiac hypertrophy. The relative cGAMP concentration in cardiac tissues collected from NAP1L1 p.D349E AAV-treated mice was upregulated compared with that in the wild-type group before and after Ang II treatment (Fig. 3K). RT–qPCR showed that Ifna and Ifnb were significantly higher in the cardiac tissues from NAP1L1 p.D349E AAV-treated mice than in the wild-type group (Fig. 3L). Through western blotting, we confirmed that Ifna and Ifnb were significantly upregulated in the MT group (Fig. 3M, N). We also used Elisa to assess the Ifna and Ifnb proteins (Supplementary Fig. S6H). Phosphorylation of Sting, Tbk1, and Irf3 increased in the MT (mutant type, Nap1l1 p.D349E) group while total Sting, Tbk1, and Irf3 did not change (Fig. 3M, N and Supplementary Fig. S6I). These results supported the fact that NAP1L1 p.D349E promotes cardiac hypertrophy through cGAS-Sting-IFN signaling.

In order to avoid the confounding factor of AngII and basal expression of Nap1l1, we generated Nap1l1 p.D349E knock-in (KI) mice (Supplementary Fig. S7A, B). Nap1l1 p.D349E KI mice exhibited significant enlargement in heart size and cardiomyocyte size (Supplementary Fig. S7C, D). Nap1l1 p.D349E KI mice also showed more severe myocardial fibrosis (Supplementary Fig. S7E, F) Nap1l1 p.D349E KI mice exhibited worse cardiac function, including LVPW;d, LVPW;s, EF, FS (Supplementary Fig. S7G, H). Correspondingly, Ifna, Ifnb, and immune factors Il-6 and Il-1b were increased in Nap1l1 p.D349E KI mice (Supplementary Fig. S7I).

## Nap1l1 p.D349E triggered cGAS-Sting-IFN signaling by weakening the oligomerization with the histone

To further investigate how Nap1l1 p.D349E triggering cGAS-Sting-IFN signaling, NRCMs were transfected with CMV-*Nap1l1* p.D349E-Flag-EGFP and CMV-*Nap1l1*-HA-EGFP plasmids. RNA-Seq also supported that the response to IFN signaling was activated in the Nap1l1 p.D349E group. Compared with the WT (wild type) group, the expression of ISGs in the MT group was increased (Supplementary Fig. S8A).

Nap1l1 p.D349E weakened the oligomerization between Nap1l1 and the histone (Fig. 4A, B). As a result, nucleosome assembly was weakened in Nap1l1 p.D349E group (Fig. 4C). We used the DNA double-strand break marker gamma-H2AX to assess DNA damage and leakage caused by Nap1l1 p.D349E. We analyzed cGAS and γ-H2ax in nuclear and cytoplasmic subfractions, and the result showed that Nap1l1 p.D349E caused the increase of cGAS and γ-H2ax in both the nucleus and cytosol (Fig. 4D–G and Supplementary Figs. S8B and S7C). To exclude the possibility that dsDNA in the cytosol is caused by mitochondrial damage, we investigated the source of cytosolic DNA. The results showed that the increase in cytosolic DNA was composed of nuclear DNA (Fig. 4H). We also examined mitochondrial transcription

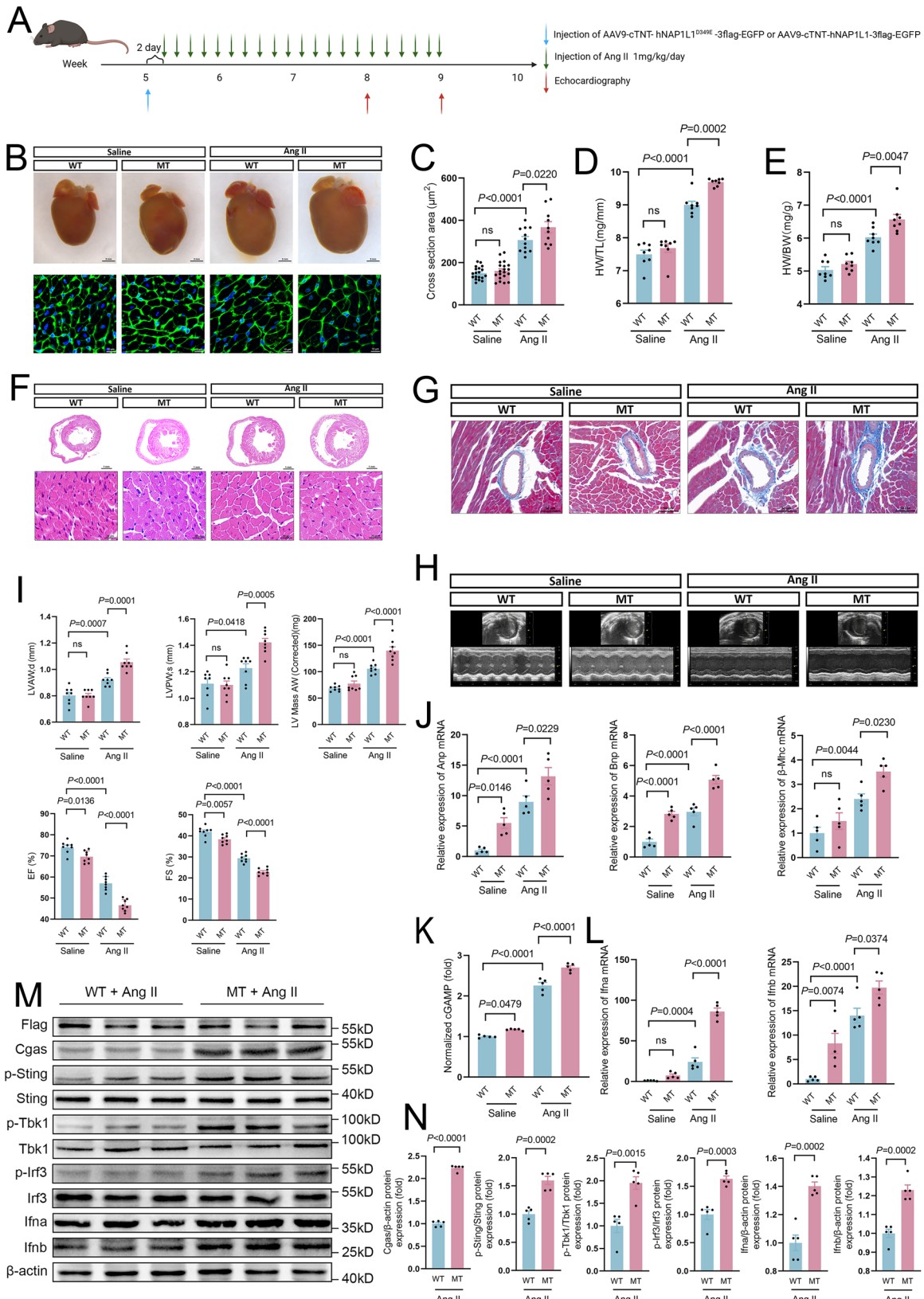

factor A (Tfam) protein levels in the WT and MT group. There was no difference between the cytoplasmic Tfam protein levels after transfection with wild-type and mutant plasmids (Supplementary Figs. S8D and S7E). Fluorescent staining of mitochondria and Tfam in cells transfected with WT and MT plasmids showed that the mitochondrial morphology basically maintained a rod-like structure, and

most of the Tfam still existed in the mitochondrial network, with very few overflowing, but between the two groups, there was no obvious difference (Supplementary Fig. S8F).

Cytosolic dsDNA is thought to be the direct activator of cGAS-Sting signaling[35]. Correspondingly, the relative concentration of the second messenger cGAMP also increased (Fig. 4I). Similar to *Nap1l1*

**Fig. 3 | Nap1l1 p.D349E exacerbated cardiac hypertrophy via cGas-Sting signaling in mouse with Ang II treatment. A** Schematic diagram depicting the experimental strategy used in C57BL6/N mice. Created in BioRender. Lv, C. (2025) https://BioRender.com/w55g327. **B, C** Representative image of heart and wheat germ agglutinin (WGA) staining and quantification of the WT, MT, WT with Ang II and MT with Ang II groups. $N = 19$, 20, 12, and 10 for WT, MT, WT with Ang II and MT with Ang II groups, and one-way ANOVA with Tukey tests were used for correction of multiple comparisons; **D** Heart weight-to-body weight (HW/BW) ratios of the WT, MT, WT with Ang II and MT with Ang II groups. $N = 8$ for each group, and one-way ANOVA with Tukey tests was used for correction of multiple comparisons; **E** Heart weight-to-tibia length (HW/BW) ratio in the WT, MT, WT with Ang II and MT with Ang II groups. $N = 8$ for each group, and one-way ANOVA with Tukey tests was used for correction of multiple comparisons; **F** Representative hematoxylin and eosin (H&E) staining showing preserved heart size in the WT, MT, WT with Ang II and MT with Ang II groups. **G** Representative images of interstitial fibrosis stained by Masson's trichrome staining showing preserved heart size in the WT, MT, WT with Ang II, and MT with Ang II groups.

**H** Representative M-mode echocardiograms of the left ventricle in the WT, MT, WT with Ang II, and MT with Ang II groups. **I** Quantification of left ventricular end-diastolic anterior wall thickness (LVAW; d), left ventricular end-diastolic posterior wall thickness (LVPW; d), ejection fraction (EF), fractional shortening (FS) or left ventricular (LV) mass; **J** qPCR analysis of Anp, Bnp, and β-Mhc in the WT, MT, WT with Ang II and MT with Ang II groups. $N = 5$ for each group, and one-way ANOVA with Tukey tests was used for correction of multiple comparisons; **K** Relative concentration of second messenger cGAMP. $N = 5$ for each group, and one-way ANOVA with Tukey tests was used for correction of multiple comparisons; **L** qPCR analysis of Ifna and Ifnb in the WT, MT, WT with Ang II, and MT with Ang II groups. $N = 5$ for each group, and one-way ANOVA with Tukey tests were used for correction of multiple comparisons; **M, N** Western blot images and quantification showing cGAS, Sting, Tbk1, Ifna, and Ifnb expression and phosphorylation of Sting, Tbk1, and Irf3 in the WT with Ang II and MT with Ang II groups. MT: mutation type, *NAP1L1* p.D349E. $N = 5$ for each group and one-way ANOVA with Tukey tests was used for the correction of multiple comparisons. Data are presented as mean values +/− SEM.

knockdown, MT also showed attenuation of heart rate and diastolic function after stress-induced elevation (Supplementary Fig. S8G). Through phalloidin staining, we found that Nap1l1 p.D349E caused a significant increase in the cell surface area (CSA) of cardiomyocytes, which was consistent with the cardiac hypertrophy phenotype of HCM patients (Fig. 4J).

Phosphorylation of Sting, Tbk1, and Irf3 was upregulated, while total Sting, Tbk1, and Irf3 levels did not change (Fig. 4K, L and Supplementary Fig. S8H). We used qPCR to assess the expression of interferon signaling molecules downstream of cGAS-Sting signaling. The expression of Ifna, Ifnb, Il-6, and Il-1b was also significantly upregulated in the Nap1l1 p.D349E group (Fig. 4M and Supplementary Fig. S8I). We cultured wild-type cardiomyocytes and cardiac fibroblasts ina culture medium after culturing the WT or MT-transfected cardiomyocytes for 48 h to simulate the in vivo immune environment. RT-qPCR revealed that both cardiomyocytes and cardiac fibroblasts respond to IFN and upregulate the expression of ISGs (Fig. 4N, O). For validation, we also performed a trans-well assay with membrane separation of WT and mutant-transfected NRCMs. The results of qPCR supported that somatic mutation Nap1l1 p.D349E triggers the IFN response of neighbor cells (Fig. 4P). We speculated that the acquisition of NAP1L1 p.D349E in some cardiomyocytes would lead to the formation of a local microenvironment of immune activation, affecting nearby cardiomyocytes and cardiac fibroblasts.

### Pharmacological inhibition of cGAS-STING-IFN signaling prevented NAP1L1 p.D349E-mediated cardiac hypertrophy

Having identified the mechanism by which NAP1L1 p.D349E promotes cardiac hypertrophy by triggering cGAS-STING-IFN signaling, we investigated whether cardiac hypertrophy could be prevented by pharmacologically inhibiting cGAS-STING-IFN signaling in mice through the use of the selective STING antagonist C-176 or the IFNAR1 antibody HY-P99137 (Fig. 5A).

Compared with those in the control mice, both C-176 and IFNAR1 antibody treatment prevented Nap1l1 p.D349E-mediated cardiac hypertrophy and fibrosis (Fig. 5B, C). The increase in the HW/TL and HW/BW ratios mediated by Nap1l1 p.D349E was prevented by C-176 and IFNAR1 antibody treatment (Fig. 5D). Echocardiography showed that C-176 and the IFNAR1 antibody ameliorated the Nap1l1 p.D349E-mediated impairment of cardiac function (Fig. 5E, F). Mechanistically, C-176 inhibited the phosphorylation of Sting and blocked the increase in Ifnb by preventing the phosphorylation of IRF3 and TBK1 (Fig. 5G, H). The expression of Sting, Tbk1, and Irf3 did not change (Fig. 5G and Supplementary Fig. S9A).

We then evaluated the safety of pharmacological inhibition of cGAS-STING-IFN signaling in animal models. In unaffected cardiomyocytes, the cGAS-STING pathway is naturally suppressed[36].

Consequently, there was no significant benefit observed from C-176 or anti-IFNAR1 antibodies in baseline mice. No abnormalities on echocardiography or cardiomyocyte abnormalities, including hypertrophy, were observed (Supplementary Fig. S9B–S9H).

## Discussion

Our study demonstrated that the NAP1L1 p.D349E variant plays a significant role in the development of HCM. While angiotensin II infusion is commonly used to model hypertension-induced cardiac hypertrophy, it is important to recognize that this model does not perfectly replicate the mechanisms underlying genetic forms of HCM. In particular, sarcomeric mutations, which are central to most genetic forms of HCM, result in distinct cellular hypertrophy pathways[37]. NAP1L1 p.D349E has been identified in COSMIC with Genomic Mutation ID COSV99673941, further implicating that this mutation could be involved in pathogenicity as a somatic mutation.

We hypothesize that the NAP1L1 p.D349E mutation contributes to a subset of myocardial cells that are less capable of tolerating normal physiological load. This hypothesis is supported by clinical observations that patients carrying the NAP1L1 p.D349E, despite lacking hypertension or other abnormal loading conditions, present with hypertrophic changes. In this context, even normal pressure loads may lead to gradual hypertrophy over time due to an inherent susceptibility of the mutated cardiomyocytes.

However, due to the limitations of replicating the long-term effects of chronic normal load in animal models, we used angiotensin II to induce short-term pressure overload as a proxy for prolonged normal load. While this approach resulted in higher transduction efficiency and more pronounced hypertrophy in mice, it simulates the chronic stress that human cardiomyocytes with the NAP1L1 p.D349E variant may experience over the years. This adaptation, though imperfect, allows us to better model the mutation's long-term effects.

Our findings suggest that the NAP1L1 p.D349E variant may predispose individuals to hypertrophy through a mechanism distinct from traditional sarcomeric mutations, potentially by altering cellular tolerance to normal physiological stress. This hypothesis aligns with our exclusion of patients with hypertension and other abnormal loading conditions, ensuring that our results reflect the intrinsic effects of the mutation rather than secondary causes of hypertrophy.

The role of innate immunity in the invasion of nonexternal DNA has attracted increasing attention. In fact, DNA that should not accumulate in the cytoplasm could be considered "foreign materials", including mtDNA and dsDNA leakage caused by DNA damage[38]. As described previously, the innate immune pathway represented by cGAS-STING signaling plays an important role in cardiac dysfunction. However, most studies involved in innate immunity in cardiac disorders have focused on mtDNA leakage. This study is the first to focus

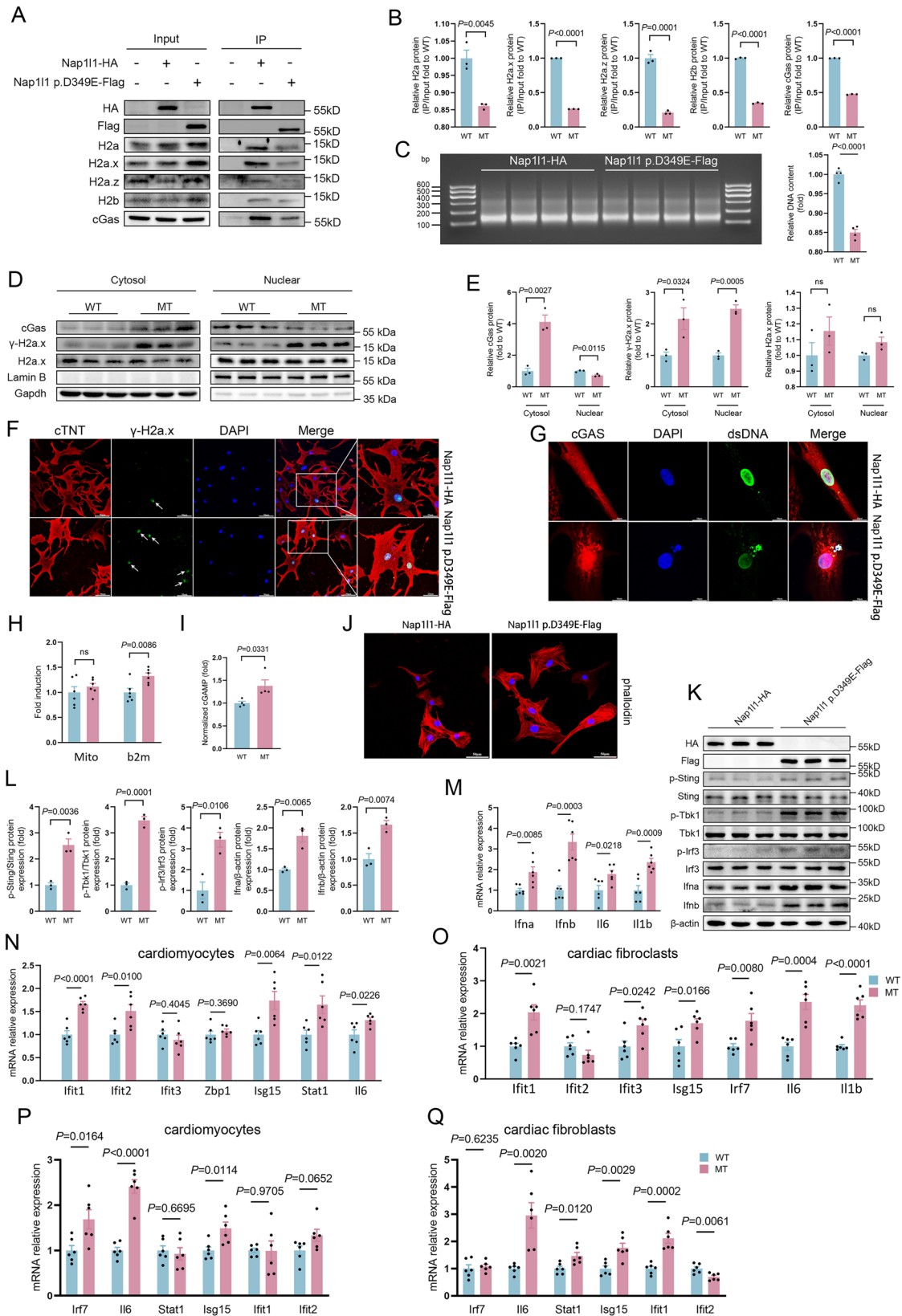

on nuclear DNA as the initiating factor to activate innate immunity and cause primary cardiac hypertrophy. Germline mutations in HCM have been extensively studied in GWAS, so we focused on somatic variants occurring locally in the left ventricle[39–41]. We selected patients with sporadic and isolated HCM to limit inheritance factors and syndromic factors. In fact, doxorubicin-induced mtDNA damage could also be

considered somatic because doxorubicin could only affect a subset of cardiac cells.

*NAP1L1* is a member of the NAP1L family, which was reported to be involved in nucleosome assembly, exchange of histone H2A-H2B dimers, transcriptional regulation, and cell proliferation[23–25]. The role of *NAP1L1* in HCM has rarely been discussed. NAP1-like is involved in

**Fig. 4 | Nap1l1 p.D349E activated cGas-Sting-IFN signaling via impaired histone oligomerization. A**, **B** Co-IP of the Nap1l1 p.D349E or wild type with H2a, H2a.x, H2a.z, and H2b; The antibodies used for pulling were IgG, HA, and Flag, and IgG is the isotype control group. $N = 3$ for each group, and two-tailed unpaired Student's *t*-test was applied; **C** Agarose gel electrophoresis showing the nucleosome-bound DNA content, representing the number of nucleosomes in the WT and MT group. $N = 4$ for each group, and two-tailed unpaired Student's *t*-test was applied; **D**, **E**, Western blot, qPCR analysis, and quantification of the levels of the γ-H2ax protein. $N = 3$ for each group, and two-tailed unpaired Student's *t*-test was applied; **F**, **G** Representative immunofluorescence staining showing the levels of γ-H2ax, cGAS, and dsDNA; **H**, **I** Relative concentration of second messenger cGAMP. For **H**, $n = 6$ and for **I**, $n = 4$, and two-tailed unpaired Student's *t*-test were applied; **J** Representative immunofluorescence staining showing the levels of phalloidin; **K**, **L** Representative Western blot images and quantification showing cGAS, Sting, Tbk1, Ifna, and Ifnb, and phosphorylation of Sting, Tbk1 and Irf3 expression in the wild-type and Nap1l1 p.D349E groups. $N = 3$ for each group, and two-tailed unpaired Student's *t*-test was applied; **M** qPCR analysis of *Ifna, Ifnb, Il-6, Il-1b*. $N = 6$ for each group, and two-tailed unpaired Student's *t*-test was applied; **N** qPCR analysis of *Ifit1, Ifit2, Ifi3, Zbp1, Isg15, Stat1*, and *Il6* in influenced cardiomyocytes. $N = 6$ for each group, and two-tailed unpaired Student's *t*-test was applied; **O** qPCR analysis of *Ifit1, Ifit2, Ifi3, Isg15, Irf7, Il6*, and *Il1b* in influenced cardiac fibroblasts. $N = 6$ for each group, and two-tailed unpaired Student's *t*-test was applied; **P**, **Q** qPCR analysis of *Irf7, Il6, Stat1, Isg15, Ifit1*, and *Ifit2* in influenced cardiomyocytes and cardiac fibroblasts. $N = 6$ for each group and a two-tailed unpaired Student's *t*-test was applied. Data are presented as mean values +/− SEM.

efficient transcription-coupled DNA repair[26,27]. Histone density is maintained during transcription by remodeling the structure of chromatin and NAP1 in vitro[28]. We speculated that NAP1L1 dysfunction may interfere with the maintenance of DNA integrity during transcription, which may leak DNA to the cytoplasm and trigger the innate immune system. Liu *et al* found that *NAP1L1* increased in the hearts of HCM patients and could regulate cardiac fibrosis through the TGF-β/Smad signaling pathway in vitro[42]. Shan *et al* reported NAP1L1 as a regulator of cardiac fibrosis, particularly in ischemic cardiomyopathy, through inhibition of YAP1 ubiquitination and degradation in cardiac fibroblasts[43]. This raises the possibility that the NAP1L1 p.D349E variant observed in our HCM cohort may also exert effects in cardiac fibroblasts, contributing to the development of fibrosis and the hypertrophic phenotype. While our study primarily focuses on cardiomyocytes, further investigation into the role of this variant in fibroblast biology could reveal additional mechanisms by which NAP1L1 p.D349E contributes to HCM pathogenesis. Given that fibrosis is a common feature in both ischemic and hypertrophic cardiomyopathy, the interaction of NAP1L1 with pathways regulating cardiac remodeling, such as YAP1, may be relevant. Future studies exploring the role of NAP1L1 in cardiac fibroblasts, as well as the crosstalk between cardiomyocytes and fibroblasts, could provide valuable insights into the broader impact of the NAP1L1 p.D349E variant in HCM and other forms of cardiomyopathy. As a nucleosome assembly protein, NAP1L1 is critical for nucleosome assembly[30]. D349 is located at the C-terminal acidic domain of NAP1L1, which is conserved in the NAP1, NAP1-like, and NAP1-related protein families. Although the C-terminal acidic domain does not mediate the direct interaction between NAP1L1 and histones, it acts as a binding assistant to contribute to oligomerization[44,45]. We also experimentally proved that p.D349E could weaken the oligomerization between Nap1l1 and histones, so we speculated that *NAP1L1* p.D349E might promote DNA damage by affecting histone deaggregation and reassembly during transcription[27,30] (Fig. 5I).

Our study demonstrates the protective effect of cGAS-STING pathway inhibition on hypertrophy in NAP1L1 p.D349E-overexpressing mice subjected to angiotensin II-induced pressure overload. To address the potential impact of these inhibitors on WT hearts, we conducted additional experiments using C-176 and anti-IFNAR1 antibodies in baseline mice. In unaffected cardiomyocytes, the cGAS-STING pathway is naturally suppressed, and we observed no significant benefit from pathway inhibition at baseline mice[36]. No abnormality on echocardiography or cardiomyocyte abnormalities, including hypertrophy, were observed. These findings provide further support for the cGAS-STING pathway as a therapeutic target in hypertrophic cardiomyopathy.

A major dilemma in the study of the pathogenic mechanism of somatic mutations is how somatic mutations with a low VAF affect the function of tissues containing trillions of cells. Here, we propose a novel mechanism with a cascade effect in which cardiomyocytes with *NAP1L1* p.D349E act as seeds to interfere with cells without *NAP1L1* p.D349E. A small number of cardiomyocytes acquire *NAP1L1* p.D349E and release type I interferon to the intercellular matrix via cGAS-STING-IFN signaling. Neighboring cells, including cardiomyocytes and cardiac fibroblasts, express interferon receptors[46]. In response to excess interferon in the matrix, cells without *NAP1L1* p.D349E and cardiomyocytes with *NAP1L1* p.D349E cooperate to promote cardiac hypertrophy (Fig. 5I).

One limitation of this study is the potential selection bias introduced by the exclusive use of HCM patients who underwent myomectomy. This cohort represents a subset of HCM patients who have more severe forms of the disease, requiring surgical intervention. Consequently, our findings may not fully capture the broader spectrum of HCM patients, particularly those with milder or asymptomatic presentations who do not undergo surgery. While the availability of myocardial tissue from myomectomy is crucial for detailed molecular and genetic analysis, it inevitably narrows the generalizability of our results. Future studies with more diverse HCM cohorts, including postmortem tissue collection from myomectomy-free patients, will be necessary to validate these findings across the entire HCM patient population. Secondly, due to the challenge of replicating the impact of low-frequency somatic mutations in myocardial tissue over the course of several years decades in the lifespan of mice, we introduced a model with high mutation load and overexpression. This approach was necessary to recreate the phenotype of cardiac hypertrophy within a feasible experimental timeframe. However, it may not fully capture the slower, progressive nature of the disease observed in human patients.

## Methods

### Study population

Seventy-one sporadic HCM patients who needed surgical intervention due to left ventricular outflow tract obstruction were recruited from Fuwai Hospital from 2012 to 2015 as the discovery cohort. This cohort was a subgroup cohort of a clinical study containing 239 HCM patients, in which sporadic cases were selected to minimize the contribution of HCM-associated germline variants. A replication cohort comprising forty-nine sporadic genetically unexplained HCM patients was recruited from the First Affiliated Hospital of Xi'an Jiaotong University. The replication cohort was a subgroup cohort of HCM patients in which 100x whole-exome sequencing (WES) in peripheral blood was performed. We recruited sporadic genetically unexplained patients in this cohort based on 100x WES data. All of the patients had outflow tract obstructions and underwent septal myectomy. None of the patients' first-degree relatives had HCM. The study was governed under the most recent (2007–2008) version of the World Medical Association's Declaration of Helsinki, and the study protocol was reviewed and approved by the ethical committee of Fuwai Hospital. All participants in this study provided written informed consent for the use of their personal health data and genetic information for research purposes. We have also obtained consent to publish potentially identifiable clinical information. Data were anonymized in accordance with ethical guidelines to protect participants' privacy. The study protocol was

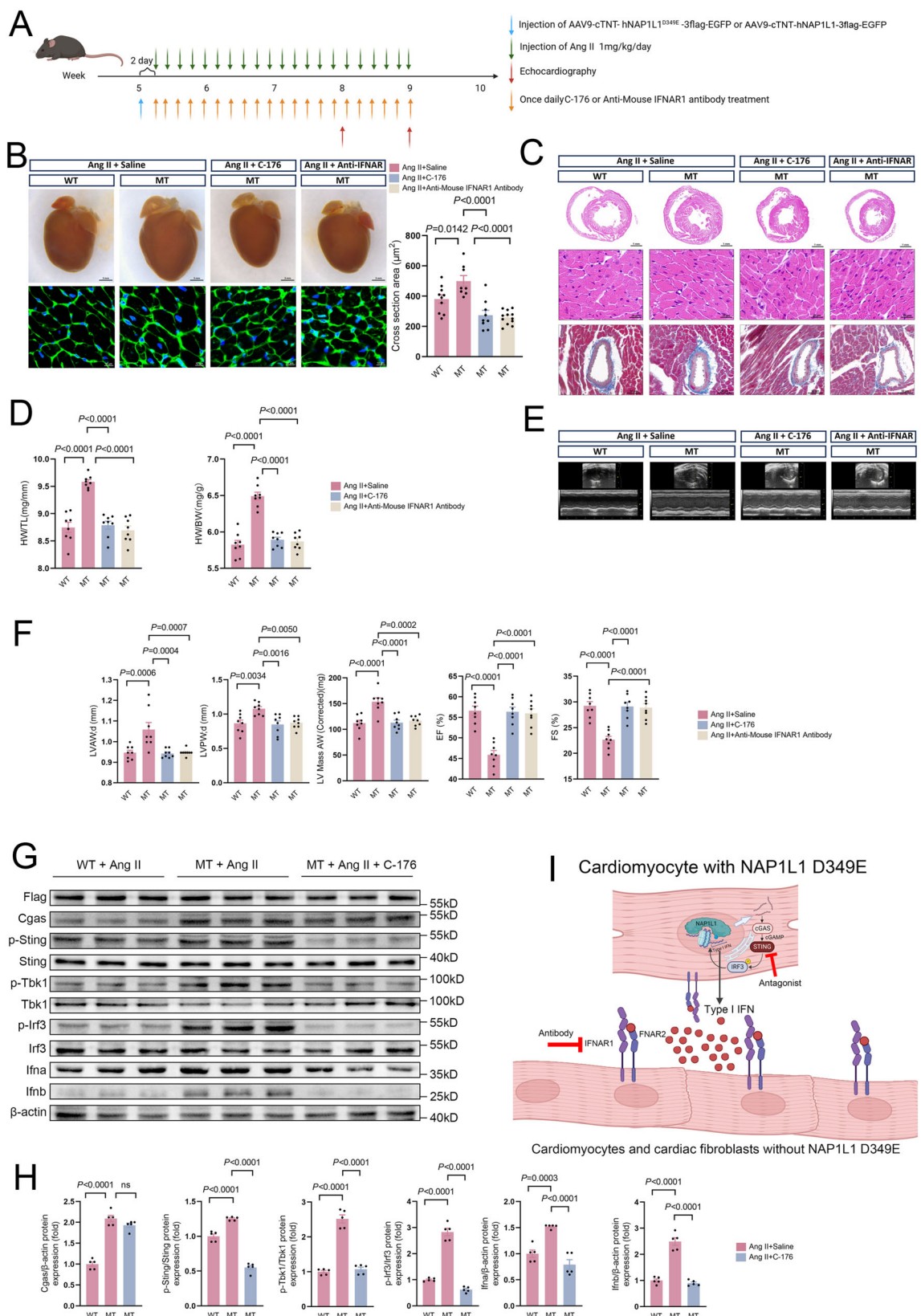

approved by Fuwai Hospital, ensuring compliance with relevant legal and ethical standards. All the patients from Fuwai Hospital and the First Affiliated Hospital of Xi'an Jiaotong University were of East Asian ancestry. The diagnosis of HCM was based on echocardiography and defined as unexplained left ventricle (LV) hypertrophy (maximum wall thickness ≥15 mm). We excluded patients with hypertension, valvular

disease, congenital disease, infiltrative cardiomyopathy, or other diseases that could lead to abnormal loading conditions.

We recruited twenty-seven septal heart transplant recipients and deceased organ donors as controls.

Sex and gender were not explicitly considered in the study design, as no significant correlation was observed between sex and the

**Fig. 5 | cGas-Sting-IFN blockade alleviated NAP1L1 p.D349E-induced cardiac hypertrophy. A** Schematic diagram depicting the experimental strategy used in C57BL6/N mice with antagonists and antibodies. C-176, a selective STING antagonist. Created in BioRender. Lv, C. (2025) https://BioRender.com/u4ll117.
**B** Representative image of heart and wheat germ agglutinin (WGA) staining and quantification of WT with AngII, MT with AngII, MT with AngII and C-176, and MT with AngII and anti-IFNAR groups; $N = 10, 8, 9, 12$ for WT with AngII, MT with AngII, MT with AngII and C-176, and MT with AngII and anti-IFNAR and one-way ANOVA with Tukey tests was used for correction of multiple comparisons; **C** Representative hematoxylin and eosin (H&E) staining and Masson's trichrome staining showing heart size, cardiomyocyte size, and fibrosis in WT with AngII, MT with AngII, MT with AngII and C-176, and MT with AngII and anti-IFNAR groups; **D** Heart weight to body weight (HW/BW) ratio in WT with AngII, MT with AngII, MT with AngII and C-176, and MT with AngII and anti-IFNAR groups. $N = 8$ for each group, and one-way ANOVA with Tukey tests were used for correction of multiple comparisons;

**E** Representative M-mode echocardiograms of the left ventricle in WT with AngII, MT with AngII, MT with AngII and C-176, and MT with AngII and anti-IFNAR groups; **F** Quantification left ventricular end-diastolic anterior wall thickness (LVAW;d), left ventricular end-diastolic posterior wall thickness (LVPW;d), Ejection fraction (EF), Fractional shortening (FS) or left ventricular (LV) mass. $N = 8$ for each group, and one-way ANOVA with Tukey tests were used for correction of multiple comparisons; **G** Western blot analysis showing the cGAS, Sting, Tbk1, Ifna, and Ifnb, and phosphorylation of Sting, Tbk1, and Irf3 expression in WT with AngII, MT with AngII, MT with AngII and C-176 groups; **H** Representative Western blot images and quantification showing the cGAS, Sting, Tbk1, Ifna, and Ifnb, and phosphorylation of Sting, Tbk1 and Irf3 expression in WT with AngII, MT with AngII, MT with AngII and C-176 groups. MT: mutation type, NAP1L1 p.D349E. $N = 5$ for each group, and one-way ANOVA with Tukey tests was used for correction of multiple comparisons; **I** Central Illustration of this study. Created in BioRender. Lv, C. (2025) https://BioRender.com/g82u862. Data are presented as mean values +/− SEM.

presence or absence of known mutations in the hypertrophic cardiomyopathy (HCM) cohort. Participants were initially grouped based on whether they carried known mutations associated with HCM. Among those without known mutations, further mutational analysis identified individuals carrying the NAP1L1 p.D349E variant, which was not previously recognized as a known mutation. Given that sex showed no association with mutation status (including the NAP1L1 p.D349E variant), sex or gender analysis was not conducted in this study.

## Whole-exome sequencing, variant calling, and germline variant classification

DNA was extracted from peripheral blood, and myocardial tissue was dissected from the left ventricle in septal myectomy. DNA from peripheral blood was subjected to WES at an average depth of 100x on an Illumina HiSeq platform. Variant calling was performed according to GATK Best Practices, and variants were annotated by ANNOVAR[47,48]. We used hg19 as the reference genome. The pathogenicity of variants in HCM-associated genes was classified as a pathogenic, likely pathogenic, variant of uncertain significance (VUS), or likely benign/benign following guidelines released by the American College of Medical Genetics and Genomics and the Association for Molecular Pathology[49].

## Somatic variant calling and deleteriousness classification

In the somatic variant analysis, we excluded patients who (1) had other diseases that could cause similar phenotypes, (2) had VUSs for their sarcomere genes, or (3) had pathogenic/likely pathogenic (P/LP) variants in 58 known genes associated with both nonsyndromic and syndromic HCM. The 58 genes related to HCM are shown in Supplementary Table S4.

DNA extracted from myocardial tissue from the septal myectomy of the patients included in somatic variant identification was subjected to WES at an average depth of 300x on an Illumina HiSeq platform. Somatic variants were identified by comparison of GATK Mutect2 with peripheral blood to remove germline variants. Variants were excluded if the maximum population minor allele frequency (MAF) was ≥0.01 for each race in the Genome Aggregation Database, Exome Aggregation Consortium, 1000 Genomes, and NHLBI Exome Sequencing Project. Hg19 was also used as the reference genome.

Established pathogenicity classes for germline mutations do not apply to somatic mutations. We considered variants to be deleterious if the CADD score was ≥20 and simultaneously predicted to be pathogenic by at least two of the algorithms, SIFT, PolyPhen2, and MutationTaster. Variants were also included when predicted to be potentially loss-of-function (LoF) (causing a stop-gain, stop-loss, start-loss, frameshift substitution, or splicing change). All variants were viewed by IGV[50]. During the manual review of variants in IGV, we used the following criteria to further filter variants: Read depth: Variants were included only if they had a minimum read depth of 20 reads to ensure sufficient coverage. Strand bias: We excluded variants that

demonstrated significant strand bias, which we defined as a ratio of variant-supporting reads on one strand being more than 70% compared to the other. Base quality: Variants were reviewed for base quality, and those with an average Phred score of <20 were excluded from further analysis. Mapping quality: Only variants with a mapping quality of >30 were considered reliable and included in the final analysis. Visual confirmation: We manually inspected the variant positions in IGV to confirm the presence of the variant across multiple reads, ensuring consistent alignment and avoiding artifacts from sequencing or mapping errors.

## Droplet digital polymerase chain reaction

To validate the WES results, NAP1L1 p.D349E was screened for a lower variant allele frequency (VAF), and droplet digital polymerase chain reaction (ddPCR) was performed for both cohorts. The detection of NAP1L1 p.D349E was performed on a QX200 ddPCR system (Bio-Rad). The primers and probes used were customized and synthesized at Integrated DNA Technologies with the following sequences: forward primer 5′-TCTTTGCAGGATTCAAATTCGC-3′, reverse primer 5′-GGTCACTTTTTACGTGAGCGT-3′, and wt probe: 5′-TCATCAT-CATCTTCAATAGC-3′ mut probe: 5′-TCATCATCATCTTCAATAGC-3′. Each reaction was set up containing fifty ng genomic DNA, nine pmol of each primer, five pmol of each probe, and 10 mL of 2x ddPCR Supermix for probes in a 20 mL reaction volume. The following PCR conditions were used: (i) an initial activation step at 95 °C for 10 min; (ii) 45 cycles of denaturation at 94 °C for 30 s and annealing/elongation at 60 °C for 1 min; and (iii) a final elongation at 60 °C for 5 min. Each reaction was set up following the manufacturer's instructions and contained 50 ng of genomic DNA. PCR was carried out following the manufacturer's instructions for each commercial assay. The PCR products were then subjected to analysis by a QX-200 droplet reader and QuantaSoft Analysis Software (Bio-Rad).

We commissioned Nanjing Geneseeq Technology Inc. to perform ddPCR in both the discovery and replication cohorts. We tested the ddPCR system in negative controls and no-template controls in different ways, and the probe always presented a clean background. Some ddPCR probes always have false positive droplets as the result of autohydrolysis or off-target effects, while other ddPCR probes may have no false positive droplets at all through multiple intrabatch and interbatch replicates. The ddPCR presented in our study was the latter case. NPA1L1 p.D349 is located in a repetitive region, i.e., Glu348Asp349Asp350Asp351Asp352Asp35, encoded by the nucleotides 5′-GAA GAT GAT GAT GAT GAT-3′. DDPCR in repetitive regions could lead to errors during PCR amplification. The WES reads were approximately 150 bp and could overlay repeat areas, so we also performed WES in 10 patients in the replication cohort, and the WES results demonstrated that ddPCR was authentic. The primer and probe sequences described in the methods could avoid PCR and/or sequencing artifacts.

## Amplicon sequencing validation

We performed amplicon sequencing on a subset of samples. To enhance detection sensitivity for low-frequency variants, we cloned the PCR products into plasmids. This approach allows the separation of individual DNA molecules, thus reducing the risk of the low-frequency variant being masked by the wild-type sequence. We then performed Sanger sequencing on multiple clones to detect the presence of the variant allele. This method allowed us to capture even low-frequency mutations at the target site. When there are more than 3 strains in the sample that are mutant by Sanger sequencing, it is considered a true positive.

## Validation of ddPCR sensitivity

To validate the sensitivity of ddPCR for detecting low-frequency somatic variants, we generated a series of plasmid mixtures containing the target variant at defined variant allele frequencies (VAFs) of 0.03%, 0.04%, 0.05%, 0.1%, 0.5%, 1%, 2%, 5%, and 10%. These samples were prepared by mixing a plasmid carrying the variant allele with a wild-type plasmid at corresponding ratios. Each sample was analyzed using ddPCR to quantify the observed VAF and assess the assay's sensitivity and accuracy. A threshold of at least three positive droplets was used to define the detection limit.

## Fluorescence-activated cell sorting

We applied fluorescence-activated cell sorting (FACS) and ddPCR to experimentally locate the variant NAP1L1 p.D349E. The variant frequencies of the somatic variant in cardiomyocytes and non-cardiomyocytes from cardiac tissues from patients D6, R35, and R38 were compared. FACS was performed on a Cytek Aurora spectral flow cytometer (Cytek Biosciences). We tested our FACS 3 times for both individuals. According to our experience and previous studies, cardiomyocytes will block if passing vertically, while passing horizontally will pass (nozzle size 100 μm)[51,52]. We cleaned the nozzle when it was blocked. In FACS, cTnT+ cells (cardiomyocytes) and cTnT− cells were distinguished (Abcam ab45932).

## Animal model

All animal use and welfare protocols adhered to the National Institutes of Health Guide for the Care and Use of Laboratory Animals following a protocol reviewed and approved by the State Key Laboratory of Cardiovascular Disease, National Center for Cardiovascular Diseases (Beijing, China). Adult male C57BL6/N mice (Beijing Vital River Laboratory Animal Technology) aged 5–6 weeks were used in the study. All mice were housed under appropriate barrier conditions with free access to food and water. The housing conditions included a 12-hour light-dark cycle and social conditions.

## Recombinant adeno-associated virus and plasmid

Recombinant adeno-associated virus serotype-9 (AAV9) harboring the human NAP1L1$^{D349E}$ gene with the cardiac troponin T (cTNT) promoter (AAV9-cTNT-hNAP1L1$^{D349E}$-3flag-EGFP) and control vectors (AAV9-cTNT-hNAP1L1-3flag-EGFP) were purchased from Genechem Company (Shanghai, China).

The plasmids for rat CMV-Nap1l1$^{D349E}$-Flag-EGFP and CMV-Nap1l1-HA-EGFP and the CMV-Flag-EGFP vector were purchased from Genechem Company (Shanghai, China).

## Animal experiments

For the experiments using AAV vectors, wild-type (WT) mice (C57BL/6 N, male) aged 5–6 weeks were intracardially injected with AAV9-cTNT-hNAP1L1$^{D349E}$-3flag-EGFP or AAV9-cTNT-hNAP1L1-3flag-EGFP. AAV9-cTNT-hNAP1L1$^{D349E}$-3flag-EGFP or AAV9-cTNT-hNAP1L1-3flag-EGFP solution was diluted with PBS. A microsyringe with a 33 G needle was used for AAV9-cTNT-hNAP1L1$^{D349E}$-3flag-EGFP and AAV9-cTNT-hNAP1L1-3flag-EGFP injection around the apex of the heart. The total amount of virus injected was $9\times10^{10}$ v.g./mouse, for a total volume of 9 μL. The needle was plunged into the left ventricle intramuscularly from the apex and then injected into three locations around the apex (front, middle, and back; 3 μL for each point) from the endocardium. Transfection efficacy was detected by Western blotting and immuno-fluorescence staining. The mice were anesthetized with 1.5% isoflurane throughout the injections.

For experiments with Ang II (Sigma-Aldrich, #A9525), mice were given Ang II (1 mg/kg/day dissolved in 0.9% NaCl) once daily by subcutaneous injection for 4 weeks beginning 2 days after AAV injection.

For experiments with C-176 (Selleck Chemicals, #S6575), mice were given C-176 (750 nmol/mouse dissolved in corn oil) once daily by intraperitoneal injection for 4 weeks beginning with Ang II injection.

For experiments with an anti-mouse IFNAR1 antibody (MedChemExpress, # HY-P99137), mice were intraperitoneally injected with an anti-mouse IFNAR1 antibody (250 μg/mouse dissolved in saline) once daily for 4 weeks beginning with Ang II injection.

Hearts were collected 4 weeks after Ang II injection.

## Echocardiography

For echocardiography, adult mice were anesthetized with 1–1.5% isoflurane. Echocardiographic parameters were obtained with a 40-MHz mouse ultrasound probe (VisualSonics Vevo 2100, VisualSonics, Toronto, Canada). Two-dimensional images were recorded in parasternal short-axis projections with guided M-mode recordings at the mid-ventricular level. Left ventricular (LV) cavity size and wall thickness were measured for at least three beats from each projection. The average LV diastolic and systolic anterior wall thickness (LVAWd, LVAWs), LV diastolic and systolic posterior wall thickness (LVPWd, LVPWs), and LV diastolic and systolic internal dimensions (LVIDd, LVIDs) were measured. LV mass and functional parameters, including fractional shortening (FS), ejection fraction (EF), and LV volume, were calculated using the abovementioned primary measurements and the accompanying software.

## Histological analyses

For mice requiring histological analysis, anesthetized mice were perfused with 60 mM potassium chloride. Hearts were excised from mice, fixed in 4% formaldehyde, and embedded in paraffin. Sections of 6 μm were cut and stained with hematoxylin-eosin (HE), Masson trichrome, and iFluor 594-conjugated wheat germ agglutinin (WGA). HE staining and standard Masson trichrome staining (LEAGENE, #DC0032) were used to assess myocardial fibrosis. iFluor 594-WGA (Solarbio, #I3320) staining was performed to determine the cross-sectional area according to the manufacturer's instructions.

## Immunohistochemical staining

Immunohistochemical staining was applied to detect α-SMA and F4/80 expression in mouse heart tissues. In brief, after antigen retrieval, paraffin sections of heart tissues were incubated with a primary antibody against α-SMA or F4/80 overnight at 4 °C, followed by incubation with an HRP-conjugated secondary antibody. Images were taken with a Pannoramic SCAN (3DHISTECH, Hungary).

## Isolation of NRCMs and cardiac fibroblasts

Primary cardiomyocytes (CMs) and cardiac fibroblasts (FBs) were isolated from Sprague-Dawley rats at the age of P1d–P3d using a Neonatal Heart Dissociation Kit (Miltenyi Biotec, #130-098-373). Briefly, 62.5 μL of enzyme P, 2300 μL of buffer X, 25 μL of buffer Y, 12.5 μL of enzyme A, and 100 μL of enzyme D were added to gentleMACS C tubes (Miltenyi Biotec, #130-093-237) for 10 neonatal rat hearts. The harvested tissue was transferred into a gentleMACS C tube, and each heart was cut into small pieces. The C tube was attached onto the sleeve of the gentleMACS Octo Dissociator with Heaters (Miltenyi Biotec, #130-096-427). The gentleMACS program 37C_mr_NHDK_1 was

used. After termination of the program, the C tube from the gentle-MACS Dissociator was detached, and the sample was resuspended in a complete culture medium (DMEM supplemented with 10% FBS and 1% penicillin/streptomycin). The cell suspension was applied to a cell strainer (70 μm) and centrifuged at 600 x g for 5 min. Red blood cells were removed using a red blood cell lysis buffer. The cell suspension was subsequently centrifuged at 600 x g for 5 min. The cell pellet was resuspended in a complete culture medium. Fibroblasts and cardio-myocytes were separated using the differential adhesion method, after which the cell suspension was plated onto a 10 cm culture dish and incubated at 37 °C for 30 min. Fibroblasts attached to the dish during this time. The culture medium was collected and centrifuged at 600 x g for 5 min. The cell pellet was resuspended in a complete cul-ture medium and plated on 1% gelatin-coated plastic culture dishes at an appropriate density. BrdU was used to inhibit the growth of residual fibroblasts in cardiomyocytes. NRCMs and NRFBs were cultured for 48 h at 37 °C in a 5% CO2 incubator before use.

### Cell transfection

For gene overexpression, plasmids for rat CMV-Nap1l1$^{D349E}$-Flag-EGFP, CMV-Nap1l1-HA-EGFP, and the CMV-Flag-EGFP vector were trans-fected using Lipofectamine 3000 (Invitrogen, #L3000-015). Seventy-two hours after transfection, the cells were collected for the experiments.

For gene knockdown, *Nap1l1* and control siRNA (Shanghai Jima Pharmaceutical Technology, China) oligonucleotides were transfected into the cells at a concentration of 0.5 nmol/L with Lipofectamine RNAiMAX (Thermo Fisher Scientific). Seventy-two hours after trans-fection, the cells were collected for the experiments.

### Immunofluorescence staining

The cells were fixed with 4% PFA for 20 min, washed twice with PBS for 5 min, and then permeabilized with 0.1% Triton X-100 for 10 min at room temperature. Then, the nonspecific sites were blocked for 1 h with goat serum (ZSGB-BIO, #ZLI-9056) at room temperature. The cells were then incubated with primary antibodies at 4 °C overnight. Then, the cells were washed with PBS and incubated for 1 h at room temperature with the appropriate secondary antibodies conjugated to Alexa Fluor-488 or Alexa Fluor-594 (1:500; Invitrogen). The cells were washed again in PBS three times and then stained with DAPI (ZSGB-BIO, #ZLI-9557) to label the nuclei.

The paraffin-embedded heart sections were deparaffinized with xylene and then rehydrated in decreasing concentrations of ethanol (100%, 100%, 95%, 90%, and 80%) and water. The heart slides were subjected to antigen retrieval by microwaving in citrate solution for 10 min; then, the slides were blocked with goat serum (ZSGB-BIO, #ZLI-9056) and incubated with primary antibody overnight at 4 °C. The next day, the slides were washed in PBST three times and incubated with the appropriate fluorescent secondary antibody (1:500, Invitrogen) for 1 h at room temperature. The slides were washed in PBST three times and then stained with DAPI (ZSGB-BIO, #ZLI-9557) to label the nuclei.

Information on the primary antibodies used is given in Supple-mentary Data 4. Images were acquired using an SP8 laser confocal microscope (Leica).

### Determination of sarcomere organization and cell surface area (CSA)

The NRCMs were fixed in 4% PFA for 20 min at room temperature, washed three times in PBS for 5 min, and then permeabilized with 0.1% Triton X-100 for 15 min at room temperature. The NRCMs were stained with 100 nM Alexa Fluor™ 647-Phalloidin (Invitrogen, #A22287) con-taining 1% BSA for 30 min at room temperature. Then, the cells were washed three times in PBS for 5 min, mounted with fluorescent mounting medium with DAPI, and visualized by SP8 laser confocal microscope (Leica).

### TUNEL assay

To detect apoptosis in NRCMs or heart tissue, NRCMs or formaldehyde-fixed paraffin-embedded sections were subjected to a terminal deoxynucleotidyl transferase-mediated dUTP-biotin nick end labeling (TUNEL) assay (Beyotime, C1086) according to the manu-facturer's instructions.

### Measurement of cGAMP in cultured NRCMs or heart tissues

cGAMP levels were measured by a 2'3'-cGAMP ELISA Kit (Cayman Chemical, #501700) according to the manufacturer's instructions. Briefly, cultured HCMECs and heart tissues were lysed in M-PER™ Mammalian Protein Extraction Reagent (Thermo Fisher, #78503) and TPER™ Tissue Protein Extraction Reagent (Thermo Fisher, #78510), respectively. One hundred microliters of immunoassay buffer C, 50 μL of the sample, 50 μL of 2'3'-cGAMP-HRP tracer, and 50 μL of 2'3'-cGAMP ELISA polyclonal antiserum were added per well. The wells were incubated overnight at 4 °C and then washed five times with 300 μL of wash buffer. A total of 175 μL of TMB substrate solution was added, and the wells were incubated on an orbital at room temperature for 30 min. Then, 75 μL of HRP stop solution was added to each well, and the plate was read at a wavelength of 450 nm. The cGAMP level was normalized to the protein concentration.

### Micrococcal nuclease (MNase) assay

Mononucleosomes were prepared using the EpiScope® Nucleosome Preparation Kit (TAKARA). In brief, collected cells were lysed, and the nuclei were pelleted by centrifugation. The pelleted nuclei were then resuspended in a prepared Micrococcal Nuclease reaction mixture and incubated at 37 °C for 30 min. The reaction was terminated by adding 0.5 M EDTA, followed by the addition of Proteinase K and a subsequent incubation at 37 °C for 30 min to obtain mononucleosomes. DNA wrapped around the nucleosomes was extracted using the NucleoSpin Gel and PCR Clean-up Kit. DNA content was analyzed via 2% agarose gel electrophoresis.

### Coimmunoprecipitation (Co-IP)

NRCMs were transfected with rat CMV-Nap1l1$^{D349E}$-Flag-EGFP or CMV-Nap1l1-HA-EGFP plasmids for 72 h. Immunoprecipitation was per-formed using an immunoprecipitation kit (Proteintech, #PK10008). Briefly, whole-cell lysates were collected. Ten percent of the lysates were taken as input, and the remaining lysates were incubated with the capture antibodies (HA, Flag, or IgG) in spin columns overnight with rotation at 4 °C. On the second day, protein A-Sepharose beads were added to the spin columns and incubated for 4 h with rotation at 4 °C. Then, the captured immunocomplexes were precipitated, washed 4–5 times with washing buffer, and mixed with elution buffer following the manufacturer's instructions. The Co-IP protein samples were heated with loading buffer and separated by SDS–PAGE following the proto-cols of the western blot assays.

### Western blot

Hearts and NRCMs were homogenized and lysed in cold RIPA buffer containing 0.1 mM phenylmethylsulfonyl fluoride (PMSF) and protease inhibitors for protein extraction. Equal amounts of total proteins were mixed and dissolved in 4 x SDS/PAGE sample buffer and heated to 100 °C for 5 min. The total protein was obtained and subjected to 12–15% SDS–PAGE according to the molecular weight of the protein. The proteins were subsequently transferred to polyvinyl difluoride membranes. The membranes were blocked for 2 h in TBST containing 5% nonfat dried milk and then immunoblotted overnight at 4 °C with the appropriate primary antibodies. On the second day, after washing three times with TBST, the membranes were incubated with HRP-conjugated secondary antibodies for 1 h at room temperature. The protein bands were detected using an ECL western blotting substrate (Thermo Scientific, #34096). Images were obtained and analyzed

using Tanon 5800 Multi and ImageJ software. Information on the primary antibodies is given in Supplementary Data 4.

## Cytosolic DNA extraction and quantification
For whole-cell cytosolic DNA analysis, NRCMs were transfected with plasmids for 48 h and collected. The cytosolic fraction was extracted using a Cell Mitochondria Isolation Kit (Beyotime, C3601) according to the manufacturer's instructions. Cytosolic DNA was isolated from the cytosolic fraction using the Universal Genomic DNA Purification Mini Spin Kit (Beyotime, D0063) according to the manufacturer's instructions. Isolated cytosolic DNA (diluted to 25 ng/mL) was used as a template for qPCR analysis of the mitochondrial DNA sequence of Mito and the nuclear DNA sequence of B2m.

## Quantitative real-time PCR (qRT–PCR)
Total RNA was isolated from tissues or cells using TRIzol reagent (Invitrogen, #15596018) following the manufacturer's instructions, and RNA concentration and purity were measured using a spectrophotometer. RNA was reverse-transcribed using Hifair® III 1st Strand cDNA Synthesis SuperMix for qPCR (gDNA digester plus) (Yeryen, #11141ES60) in accordance with the manufacturer's instructions. qRT–PCR analysis was carried out using SYBR Green (Yery, #11184ES60) on a QuantStudio 5 detector (Thermo Fisher Scientific, Waltham, USA) under standard PCR conditions (95 °C for 5 min, followed by 40 cycles of 95 °C for 10 s and 60 °C for 30 s, with a final dissociation stage). The fold difference in gene expression was calculated using the $2^{-\triangle\triangle Ct}$ method and is presented relative to Gapdh mRNA. The primer sequences are given in Supplementary Data 5.

## Cytoplasmic and nuclear fraction extraction
Cytoplasmic and nuclear fractions were extracted using the Minute™ Cytoplasmic & Nuclear Extraction Kit for Cells (Invent, SC-003) according to the manufacturer's instructions. Briefly, an appropriate amount of Cytoplasmic Extraction Buffer was added to the cell pellet to lyse the cells, followed by high-speed centrifugation to separate the cytoplasmic and nuclear fractions. A nuclear extraction buffer was added to the nuclear pellet to lyse the nuclei, and the lysate was then subjected to high-speed centrifugation using a protein extraction filter cartridge to obtain the nuclear fraction.

## Monitoring of the electrophysiologic activity of NRCMs
To evaluate changes in the physiological activity of cardiomyocytes, a CardioExcyte 96 instrument (Nanion Technologies GmbH, Germany) was used to monitor the heart rate (beat rate) and contractility (amplitude) of the NRCMs. NRCMs were seeded at an appropriate density onto a CardioExcyte 96 Sensor Plate (Nanion Technologies) precoated with fibronectin, and the culture media was replaced after 24 h. After 48 h, the NRCMs exhibited stable beating. The culture medium was replaced 2 h prior to transfection with plasmids or siRNA, and NRCM activity was continuously monitored in real-time at 2-hour intervals as required.

## RNA isolation and sequencing
Total RNA was isolated from NRCMs transfected with *Nap1l1* and control siRNA followed by library preparation according to Illumina standard instructions (VAHTS Universal V6 RNA-seq Library Prep Kit for Illumina®). An Agilent 4200 bioanalyzer was used to evaluate the concentration and size distribution of the cDNA library before sequencing with an Illumina NovaSeq 6000. High-throughput sequencing was performed fully according to the manufacturer's instructions (Illumina). The raw reads were filtered by Seqtk before mapping to the genome using HISAT2 (version 2.0.4)[53]. The gene fragments were counted using stringtie (v1.3.3b) followed by TMM (trimmed mean of *M* values) normalization[54–56]. A gene set enrichment analysis (GSEA) was performed based on curated gene sets from the Broad Institute's molecular signature database MSigDB.

## Nap1l1 p.D349E knock-in mice model
The floxed Nap1l1 p.D349E mutant knockin mouse was generated using CRISPR/Cas9 technology (Cyagen, China). To generate Nap1l1 +/flox-D349E;αMHC-Cre mice, the floxed Nap1l1 p.D349E mutant mouse were cross-bred with αMHC-Cre mouse line. TAM (Sigma) was administered intraperitoneally at 100 mg per kilogram of BW daily for 7 days. Adult male C57BL6/N mice (Beijing Vital River Laboratory Animal Technology) at 5–6 weeks were used in treatment with Ang II. All mice were housed under appropriate barrier conditions with free access to food and water. Housing conditions include a 12-hour light-dark cycle and social conditions.

## Multivariate linear simulation
We employed a linear model to explore the relationship between the NYHA class variable and the predictors NAP1L1 p.D349E and duration of disease. The model was formulated as follows:

NYHA class $= \beta_0 + \beta_1 \times$ whether carrying somatic NAP1L1 p.D349E $+ \beta_2 \times$ duration of disease $+ \epsilon$

$\beta_0$ is the intercept, $\beta_1$ is the coefficient of whether carrying somatic NAP1L1 p.D349E, $\beta_2$ is the coefficient of duration of disease, and $\epsilon$ represents the error term. The predictors were selected based on their hypothesized biological relevance to the NYHA class. The duration of the disease was standardized to facilitate coefficient interpretation, and multicollinearity was assessed using variance inflation factors (VIFs), all of which were below 10. The model was fitted using ordinary least squares (OLS), with robust standard errors applied to account for potential heteroscedasticity.

## Statistics and reproducibility
All experiments were independently repeated at least three times with similar results. For representative images, such as micrographs, images were selected from a total of ten independent experiments. Data are presented as mean values +/− SEM. To evaluate differences between experimental groups, a one-way analysis of variance (ANOVA) was performed. When a significant difference was detected ($p < 0.05$), Tukey's Honest Significant Difference (Tukey HSD) test was used for post hoc multiple comparisons. The association between the New York Heart Association class and NAP1L1 p.D349E was tested by the Kendall rank correlation coefficient. The data were analyzed by using SPSS software, version 23.0 (SPSS, Inc., Chicago, IL), Prism software (version 9.4.0, GraphPad, America), and R (version 4.1.2). Representative images were chosen to show parameters close to the mean value for each group.

## Reporting summary
Further information on research design is available in the Nature Portfolio Reporting Summary linked to this article.

# Data availability
The authors confirm that data supporting the findings of this study are available in the manuscript and its supplements. The raw WES data generated in this study have been deposited in the Genome Sequence Archive (Genomics, Proteomics & Bioinformatics 2021) at the National Genomics Data Center (Nucleic Acids Res 2022), China National Center for Bioinformation/Beijing Institute of Genomics, Chinese Academy of Sciences (GSA-Human): HRA010055. The raw sequencing data are available under controlled access due to data privacy laws related to patient consent for data sharing, and the data should be used for research purposes only. Access can be obtained by approval via their respective DAC (Data Access Committees) in the GSA-human database. According to the guidelines of GSA-human, all non-profit researchers are allowed access to the data, and the Principal Investigator of any

research group is allowed to apply for controlled access to the data. For data requests, please refer to the detailed guide: https://ngdc.cncb. ac.cn/gsa-human/document/GSA-Human_Request_Guide_for_Users_ us.pdf. DAC will respond within two weeks. The data will be available within a week once the access has been granted, and they will be available to download for one year. The raw RNA dataset is available at the National Center for Biotechnology Information's Gene Expression Omnibus Database GSE286501 [https://www.ncbi.nlm.nih.gov/geo/ query/acc.cgi?acc= GSE286501]. Source data are provided in this paper.

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

## Acknowledgements

We thank all the patients and their families for participating in our study. The central illustration was created using BioRender.com. This study was supported by the Chinese Academy of Medical Sciences (Innovation Fund for Medical Science, 2023-CXGC-SYS01-2, 2021-I2M-1-016) Y.Wang, National Natural Science Foundation of China (82470450, 81970430) Y.Wang, National Key Research and Development Program of China (2017YFC0909400) Y.Wang and Basic Research Fund of State Key Laboratory of Cardiovascular Disease Y. Wang.

## Author contributions

Conceptualization: Y. Wang. Methodology and software: C.L., X.A., and F.W. Investigation, resources, and data curation: C.L., X.A., X.X., J.W., S.W., G.W., Y.Z., and Y. Wu. Writing: C.L., X.A., X.X. F.W. and Y. Wang. Revising: C.L., X.A. and Y. Wang. Assistance in data analysis and revision: Y.Wu, H.C., R.H., and L.S., Funding acquisition: Y. Wang.

## Competing interests

The authors declare no competing interests.
