## [Transparent Peer Review file · Nature Communications]

Somatic NAP1L1 p.D349E promotes cardiac hypertrophy through cGAS-STING-IFN signaling

Corresponding Author: Professor Yibo Wang

Version 0:

Reviewer comments:

Reviewer #1

(Remarks to the Author)

The work presented by Cheng Lv and their 4 other first authors and other co-authors is focused on the discovery of a potentially novel somatic variant that is associated with hypertrophic cardiomyopathy (HCM). The authors accurately cite that somatic mutations are associated with age-related disorders including cancer and cardiovascular disorders but are more difficult to study because of the difficulty in obtaining myocardial tissue. The current study leverages the unique availability of myocardial tissue from HCM patients undergoing surgical myomectomy to identify a novel somatic variant in the Nucleosome assembly protein 1 like 1 (NAP1L1) gene associated with HCM and worse outcomes than those patients with germline variants. The identified variant p.D349E in the NAP1L1 gene is conserved and predicted to cause dysfunction. Using in-vitro and in-vivo models, the authors demonstrate that NAP1L1 dysfunction is associated with cardiac hypertrophy through a proposed mechanism of activation of the cGAS-STING pathway. While this association is intriguing there are several conceptual and technical gaps that limit the support of the conclusions by the available data.

1. Characterization of the somatic mutations in the clinic cohorts – The authors use a cohort of patients with sporadic HCM to investigate causes of the non-germline HCM. While it is understood that the need for myocardial tissue necessitates surgical intervention, only using HCM patients undergoing myomectomy, does result in a form of selection bias. This is a minor concern but should be mentioned as a possible limitation. There are several additional concerns:

a. There is no description of the criteria for selection of patients undergoing myomectomy. What was the average LV outflow tract gradient? Was it higher in the D349E population? This should be included in the description of the clinical cohort.

b. Whole genome sequencing (WES) demonstrated the presence of D349E variant in the germline-negative HCM patients which was reasonable for the discovery cohort. However, in the validation cohort or even in the originally excluded patients, what was the VAF of the NAP1L1 p.D349E variant? It is possible that the presence of the somatic variant might contribute to worse clinical outcomes in those patients who also have a pathogenic or likely-pathogenic germline variant. If the VAF is less than the detection efficiency of ddPCR or amplicon-sequencing, in those patients with germline variants, there what is the mechanism for the segregation? Given the claim that 7 of 49 patients who required myomectomies (14.3%) were positive for the NAP1L1 p.D349E variant, it would seem plausible that some patients would be positive for both germline and somatic mutations. At the minimum, the VAF for NAP1L1 p.D349E by ddPCR in myocardial tissue samples should be reported for EACH PATIENT with germline mutations to determine if NAP1L1 p.D349E variant contributes to worse outcomes in patients with traditional HCM variants.

c. Digital droplet PCR (ddPCR) is a sensitive technique for identification of single variants. While it is appreciated that the authors validated some of the patients was both WES and ddPCR, there is no description of the sensitivity of the ddPCR assay for this site. The report a range of VAF% from 0.16 to 4%. A secondary validation with another assay such as amplicon sequencing which has far greater coverage than WES (often 50-100k reads per amplicon) would be important to confirm the findings

d. Identification of the NAP1L1 p.D349E variant by FACs is reasonable, although there appears to be a considerable discrepancy between the reported VAF of the cTnT+ cells and the previously reported VAF in Supplemental Table S8. What is the unfractionated sample? This is not described in the text and accounts for approximately 50% of the cTnT+ sample. What are these cells? A more detailed description of this experiment and the relative contribution of NAP1L1 p.D349E positivity from each fraction should be included.

e. Minor point – The description of the isolation of DNA from the patients is limited. How much tissue was used, were multiple sites samples, was there any gross differences in fibrosis or other structural changes between study patients.

f. Minor point – The labeling of the ddPCR graphs is very poor and confusing. Please include appropriate labels for positive signals and the source of each sample, this is especially true for the negative controls in Supplemental figure S2. It is entirely unclear the source of these samples.

2. Mouse model of NAP111 p.D349E – The authors describe animal model of NAP1L1 p. D349E to demonstrate the correlation of the presence of the somatic mutation and hypertrophic cardiomyopathy (HCM) using AAV9 expression of the variant within the heart. While it is intriguing that the expression of AAV containing NAP111 p.D349E appears to recapitulate some of the aspects of HCM there are several issues with the model in its current form.

a. Expression levels of AAV9-cTnT-hNAP111D349E are poorly characterized and do not match the clinical data. What is the transduction efficiency of the AAV9-cTnT-hNAP111D349E and AAV9-cTnT-hNAP111WT viruses? The images shown in Supplemental Figure S6 are of low resolution making it difficult to determine the percent of transduced cardiomyocytes but the efficiency of transduction is far higher than the VAF % of NAP111 p.D349E detected in humans (range 0.16-4%), making the comparison more difficult. What is the VAF % of NAP111 p.D349E in transduced animals? Is there any correlation to the VAF % to the degree of hypertrophy and ventricular dysfunction? The transduction efficiency of viral transduction should be reported for each cohort in addition to quantifying the VAF %.

b. The use of angiotensin II as a model for HCM – While the infusion of angiotensin II is a well-established model of hypertension-induced cardiac hypertrophy and ventricular dysfunction, it does not produce cellular hypertrophy in the same manner as genetic forms of HCM and therefore it is difficult to equate the two pathways. The well-characterized aMHC-R403Q knock-in mouse has been shown to have a different response to pressure overload supporting the hypothesis that sarcomeric or related genetic forms of HCM are distinct from pressure/stress induced forms of hypertrophy (PMID: 12925456). Therefore, it is difficult to determine if the model presented here with over-expression of NAP111 p.D349E in the context of pressure-overload truly represents a genetic form of HCM or is in-fact a demonstration of susceptibility to pressure-overload hypertrophy. What is the phenotype of the NAP111 p.D349E over-expressing mice without angiotensin II treatment in older mice? Many HCM mouse models do not demonstrate a phenotype until 24 weeks of age. If the claim is that patients with the NAP111 p.D349E variant present at later ages and with more severe phenotypes of hypertrophy, then a corresponding aged model should be performed.

In the human cohorts, were the affected patients hypertensive? What about patients with hypertension induced cardiac hypertrophy and/or heart failure, do they have an increased VAF % for the NAP111 p.D349E variant? Is possible that the NAP111 p.D349E variant predisposes patients to pressure-overload hypertrophy but not with the same mechanism as traditional sarcomeric forms of HCM? A discussion and explanation of this possibility along with the above data should be included.

c. The proposed model by the authors is that the presence of the NAP111 p.D349E variant even at very low VAF%'s (0.16%) induces cardiac hypertrophy through stimulation of the cGAS-STING-IFN signaling pathway leading to expression of cardiac hypertrophic-inducing cytokines. What is the minimal number of positive cardiomyocytes with the NAP111 p.D349E variant that produces the phenotype? The dose of AAV9 should be titrated down to determine the minimal VAF% that induces cardiac hypertrophy and dysfunction. Furthermore, the authors should perform in-vitro studies at low levels of transduction to validate their proposed mechanism. A trans-well assay with membrane separation of WT and mutant-transfected (MT) NRCMs at various MT transfection doses would also serve to support the hypothesis of a paracrine effect on WT cardiomyocytes induced by a small number of NAP111 p.D349E variant-containing CMs.

3. Rescue of hypertrophy with inhibition of cGAS-STING pathway – The authors demonstrate that with inhibition of the cGAS-STING using both a small-molecule inhibitor and antibody, there is prevention of cardiac hypertrophy in animals over-expressing the NAP111 p.D349E variant in the context of angiotensin II treatment (pressure overload). While these data are compelling, there is no corresponding controls for C-176 and anti-mouse IFNAR1 antibodies. What is the effect on WT hearts at baseline or exposed to angiotensin II? These data would be critical prior to any attempt of clinical translation for this work.

Minor points: There are multiple grammatical and spelling errors throughout the manuscript.

Reviewer #2

(Remarks to the Author)

In this study, the authors have discovered a new somatic variant responsible for hypertrophic cardiomyopathy (HCM). Using two patient cohorts, the authors detected that a fraction of HCM patients have a recurrent somatic variant, NAP1L1 p.D349E, specifically in cardiomyocytes. The authors have shown that knockdown (KD) of Nap111, in vitro leads to the release of nuclear dsDNA into the cytosol, which activates the cGAS-STING pathway. Ectopic expression of the D349E variant promotes HCM in mice by locally forming a highly inflamed microenvironment. Mechanistically, the authors have shown that Nap111 induces genomic instability by affecting nucleosome assembly, which leads to the release of dsDNA into the cytosol. Most importantly, the authors have shown that pharmacological inhibition of STING or IFNAR1 antibody treatment could improve conditions in mice with HCM. This is a very interesting study showing the effect of a somatic variant in HCM; however, further evidence could strengthen the findings and could yield a better understanding of how the Nap111 D349E variant causes HCM.

1. A recent study reported that NAP1L1 (PMID: 37593048) is a regulator of cardiac fibrosis and is upregulated in ischemic cardiomyopathy patient hearts and regulates cardiac fibrosis by inhibiting YAP1 ubiquitination and degradation. The authors should comment on this how this Nap111 variant may function in the cardiac fibroblasts and possibly contribute to HCM observed.
2. Similar to the first comment, it appears that cGAS interaction and expression levels were affected by D349E (fig. 3L, 4A, and 5G). Can the authors confirm whether the downstream activation of IFN signaling is due to altered cGAS stability or enhanced dsDNA release and sensing?
3. Data presented in fig. 4A alone did not support the claim that Nap111 variants affect nucleosome assembly. Did the authors find any DNA damage-related gene sets in the RNA-seq presented in fig. S5A? In addition, the authors should show the effect of the D349E variant on Nap111 expression and nucleosome assembly.
4. The authors should clearly mention in the legends of fig. 4A which antibody was used for the pull-down and include the isotype controls in the experiment.
5. The results shown in fig. 4G are not enough to support that dsDNA released to the cytosol is not from mitochondria. The authors should stain cells as shown in fig. 4F with mitochondrial markers to exclude mitochondrial damage and mtDNA release (i.e. –analysis of TFAM outside of mitochondrial networks).
6. The authors have shown that mRNA levels of cytokines are increased after Nap111 silencing or D349E expression (fig. 2I, 4L and 4N). cGAS is not a robust inducer of pro-inflammatory cytokines. Thus, this data could be strengthened if the authors can show the levels of released cytokines in culture media supernatants or tissue extracts.
7. For mouse experiments, the authors express human Nap111 variants in the presence of WT mouse Nap111. To better understand the function and effect, the authors should use genetically engineered models in their study. Can the authors knockdown/knockout endogenous Nap111 then express the D349E variant to cleanly assess the role of this protein in the absence of WT protein.
8. The authors should quantify the IF showed in fig 2A, 2B, 4E, 4F and S7B.

Reviewer #3

(Remarks to the Author)

The authors examine the role of a specific somatic variant NAP1L1 p.D349E in cardiac hypertrophy. They evaluate two different cohorts of HCM patients by typical NGS followed by variant calling to identify a novel variant, NAP1L1 as potentially causative in sporadic cases of HCM. The authors subsequently perform a series of animal experiments to recapitulate the phenotype observed in the human cohorts. The authors are attempting to answer an important question in the literature about the contribution of somatic variants in CVDs, an emerging area of inquiry given the enhanced sequencing technologies now in broad use. This line of inquiry has merit and is of great interest to the field of cardiovascular genetics.

Major

Introduction:

The contents of the introduction do not effectively convey the hypothesis to the reader. It would be improved with the addition of a hypothesis describing the putative mechanism on HCM (page 4 lines 15-17)

Results:

I am concerned that the overall approach of using ddPCR as well as the next generation sequencing techniques are not robust enough to evaluate a mutation with a frequency this low. Sanger sequencing similarly would not be an effective means for determining the absence or presence of this variant in the heart (or other) tissue. As it stands, it is not able to be determined if the variant of interest is bona-fide or artifact of the chosen assays. This is true for both the tissue and blood based approaches. Something like FISH (or mutaFISH) confirmation and/or UMI-based, deep error-corrected NGS would be necessary to confirm the presence of the variant of interest in the tissue given the low VAF in the tissue.

If the variant is indeed real, there is an extremely low VAF among patient samples in the tested tissues with the majority less than 1%. The key question to answer is whether dysfunction among ~2-8% of cells (assuming heterozygosity) in a given anatomical region is enough to cause symptomatic hypertrophy and cardiovascular dysfunction? The authors attempt to address this in the discussion by offering a paracrine/autocrine type hypothesis that is untested.

The animal models do seem to exert an effect. However, this model does not seem to accurately recapitulate the circumstances in the human physiology identified in the two cohorts. By looking at the flag staining (SF S6), it would appear that far more than 4-8% of cells are affected, making it difficult to triangulate the magnitude of the effect for a much smaller proportion of cells.

Are the authors able to further evaluate any other heart region or non-dividing tissue in their cohorts for the SNP using ddPCR to determine cardiac specificity and or intra-cardiac regional?

Have the investigators considered the effects of polyploidy in the setting of cardiac tissue? This would potentially confound the reported VAFs to be artificially higher than they actually are.

It is notable that no phenotypic changes are present in animal hearts without the addition of Ang-II. Do the authors have any insight as to why this is the case?

Minor

Nomenclature of genes and gene products – Genes in Italics and gene products in normal type. Is “NAP1-like” the gene product of NAP1L1?

No reference given for what genome build was used in alignment of the sequencing. In GRCh38, the genomic position of the mutation in question is Chr12:76050543 whereas the location shown in the BAM chr12:76444323

The SNV in question has been identified in COSMIC: <https://cancer.sanger.ac.uk/cosmic/mutation/overview?id=177063780>

Manual review of IGV was outlined in the methods but the criteria undertaken to further filter the variants were not defined.

Can density of total and small arteriolar area be quantified? (Supplementary Figure S6D)

Page 1, Line 6 – should read “somatic variants are”

Page 1, Line 13 – “contributes”

Version 1:

Reviewer comments:

Reviewer #1

(Remarks to the Author)

Overall, the revised manuscript is significantly improved and the majority of my concerns have been addressed. However the sensitivity of the primary method to detect the somatic variant, namely ddPCR, still is not adequately addressed. The authors used the traditional method of cloning the region by PCR and then performing Sanger sequencing to identify the variant. While this method demonstrates that a variant is present, it does not address the central question of validating the SENSITIVITY of the primary method of ddPCR that the authors rely on for quantification of the somatic variant. The suggestion of amplicon sequencing was to amplify the region and use NextGen Sequencing (NGS) as a secondary QUANTITATIVE method to validate the sensitivity of the ddPCR results. At the minimum, the authors should clarify how many clones they amplified and sequenced (using SANGER sequencing) to provide an estimate of the sensitivity ddPCR for the same samples. An alternative method would be to perform serial dilutions of a plasmid containing the variant with WT plasmids to generate specific populations, 5%, 2%, 1%, 0.5%, 0.1%, 0.05%. Then perform ddPCR on each sample and generate a correlation curve. While the authors cite their own papers in reference to the sensitivity of ddPCR for somatic mutations, in those papers the detection frequencies were far higher (80-90%) and therefore the low sensitivity of the assay was not addressed.

Reviewer #2

(Remarks to the Author)

Thank you for the opportunity to evaluate the revised manuscript. The authors have added additional experiments to strengthen the conclusions of the paper and address my questions. However, I still have some remaining issues. First regarding my original comment 6, I was asking about inflammatory cytokines, not simply type I interferons. The IFN ELISA data provided in response to review is convincing, but did the authors see IL-6 or IL-1 protein secretion? As mentioned, cGAS is not a strong inducer of pro-inflammatory cytokines, and IFN α /b are interferons, not inflammatory cytokines. The authors should provide additional clarification and data, as well as including the IFN ELISA data in the revised manuscript, not simply in response to reviewers. Regarding comment 7 and response, I am unclear why the authors chose not to include the data from the NAP1L1 D349E knock-in mouse model, where WT NAP1L1 is ablated. I think these data are supportive of a role for NAP1L1 mutation in hypertrophy and should be included, especially if the cGAS/STING pathway is activated and interferon-related genes are increased. They support the AAV models and do not involve the AngII confounder.

Reviewer #3

(Remarks to the Author)

No additional comments to the authors. Care and diligence were taken to address the previous concerns.

Version 2:

Reviewer comments:

Reviewer #1

(Remarks to the Author)

The current experimental data to verify the sensitivity of ddQPCR is adequate. The data is clearly presented and actually provides a nice method for confirmation of low frequency VAFs that should be highlighted.

Reviewer #2

(Remarks to the Author)

The authors have satisfactorily addressed my comments.

Full title: Somatic NAP1L1 p.D349E mediates sporadic hypertrophic cardiomyopathy,

Running title: NAP1L1 p.D349E mediates cardiac hypertrophy

Authors' names and affiliations: Cheng Lv^{1,5}, Xiayidan Alimu^{1,5}, Xiao Xiao^{1,5}, Fei Wang^{1,5}, Jizheng Wang^{1,5}, Shuiyun Wang², Guixin Wu¹, Yu Zhang¹, Yue Wu³, Houzao Chen⁴, Rutai Hui¹, Lei Song^{1,6}, Yibo Wang^{1,6,7}

¹State Key Laboratory of Cardiovascular Disease, Fuwai Hospital, National Center for Cardiovascular Diseases, Chinese Academy of Medical Sciences and Peking Union Medical College, Beijing, China.

²Department of Cardiac Surgery, Fuwai Hospital, Chinese Academy of Medical Sciences and Peking Union Medical College, Beijing, China.

³Department of Cardiovascular Medicine, The First Affiliated Hospital of Xi'an Jiaotong University, Xi'an, China.

⁴State Key Laboratory of Common Mechanism Research for Major Diseases, Department of Biochemistry and Molecular Biology, Institute of Basic Medical Sciences, Chinese Academy of Medical Sciences and Peking Union Medical College, Beijing, China.

⁵The first five authors contributed equally to this article.

⁶Correspondence: songlqd@126.com and yibowang@hotmail.com

⁷Lead contact, Yibo Wang, Ph.D., Fuwai Hospital, National Center for Cardiovascular Diseases, Chinese Academy of Medical Sciences and Peking Union Medical College, 167 Beilishi Rd, Beijing, 100037, China. E-mail: yibowang@hotmail.com, Phone number: +86-10-60866112

Reviewer #1

Comment 1: Characterization of the somatic mutations in the clinic cohorts. The authors use a cohort of patients with sporadic HCM to investigate causes of the non-germline HCM. While it is understood that the need for myocardial tissue necessitates surgical intervention, only using HCM patients undergoing myomectomy, does result in a form of selection bias. This is a minor concern but should be mentioned as a possible limitation.

Our response: We appreciate your constructive comment. The selection bias due to only using HCM patients undergoing myomectomy is indeed unavoidable, given the necessity of myocardial tissue for our investigation. We have acknowledged this limitation in the Discussion section as follows:

One limitation of our study is the potential selection bias introduced by the exclusive use of HCM patients who underwent myomectomy. This cohort represents a subset of HCM patients who have more severe forms of the disease, requiring surgical intervention. Consequently, our findings may not fully capture the broader spectrum of HCM patients, particularly those with milder or asymptomatic presentations who do not undergo surgery. While the availability of myocardial tissue from myomectomy is crucial for the detailed molecular and genetic analysis, it inevitably narrows the generalizability of our results. Future studies with more diverse HCM cohorts, including post-mortem tissue collection from myomectomy-free patients, will be necessary to validate these findings across the entire HCM patient population.

Comment a: There is no description of the criteria for selection of patients undergoing myomectomy. What was the average LV outflow tract gradient? Was it higher in the D349E population? This should be included in the description of the clinical cohort.

Our response: We thank you for highlighting this important point. The criteria for selecting patients undergoing myomectomy were based on standard clinical indications, including severe symptomatic hypertrophic obstructive cardiomyopathy (HOCM) that was refractory to medical management. In line with this, patients were primarily selected based on the presence of significant left ventricular outflow tract (LVOT)

obstruction, as assessed by echocardiography.

Regarding the left ventricular outflow tract gradient, we have now described the mean LVOT gradient of this cohort in detail in the revised manuscript. In addition, we examined whether the gradient was higher in patients with the D349E mutation, which was slightly higher than in patients without any mutation, but lower than in patients with a known germline pathogenic mutation. We have also added these findings to the clinical cohort description for completeness.

Clinical Cohort Selection:

Patients undergoing myomectomy were selected based on the presence of symptomatic HOCM with significant LVOT obstruction (typically defined as a resting gradient >50 mmHg or provokable gradient >70 mmHg) despite optimal medical therapy. The average resting LVOT gradient for the cohort was 54 ± 23 mmHg. We also evaluated the subgroup of patients carrying the D349E mutation and found that their LVOT gradients were 57 ± 21 mmHg, slightly higher than in patients without any mutation, but lower than in patients with a known germline pathogenic mutation.

We also added detailed description of the cohort in the revised manuscript:

The average resting LVOT gradient for the cohort was 54 ± 23 mmHg.

We then evaluated the subgroup of patients carrying the D349E mutation and found that their LVOT gradients were 57 ± 21 mmHg, slightly higher than in patients without any mutation, but lower than in patients with a known germline pathogenic mutation.

Comment b: Whole genome sequencing (WES) demonstrated the presence of D349E variant in the germline-negative HCM patients which was reasonable for the discovery cohort. However, in the validation cohort or even in the originally excluded patients, what was the VAF of the NAP1L1 p.D349E variant? It is possible that the presence of the somatic variant might contribute to worse clinical outcomes in those patients who also have a pathogenic or likely-pathogenic germline variant. If the VAF is less than the detection efficiency of ddPCR or amplicon-sequencing, in those patients with germline variants, there what is the mechanism for the segregation? Given the claim that 7 of 49 patients who required myomectomies (14.3%) were positive for the NAP1L1 p.D349E variant, it would seem plausible that some patients

would be positive for both germline and somatic mutations. At the minimum, the VAF for NAP1L1 p.D349E by ddPCR in myocardial tissue samples should be reported for EACH PATIENT with germline mutations to determine if NAP1L1 p.D349E variant contributes to worse outcomes in patients with traditional HCM variants.

Our response: We appreciate your concerns regarding the presence of the NAP1L1 p.D349E variant in patients with known pathogenic mutations. In our original manuscript, the term "genetically unexplained" may have led to some confusion. To clarify, our replication cohort was derived from the Xi'an Jiaotong University HCM cohort, specifically excluding patients with known pathogenic mutations. Based on our experience, somatic mutations and germline mutations may exist simultaneously and participate in pathogenicity through different mechanisms^{1,2}. It will be one of the important directions of our subsequent research to investigate how known cardiomyopathy-causing mutations and somatic mutations interact and work together to promote myocardial hypertrophy.

Regarding the detection limits of ddPCR, the method is highly sensitive, capable of detecting very low VAF, typically down to 0.01% or even lower, depending on the sample quality and assay conditions. We commissioned Nanjing Geneseq Medical Laboratory to perform the ddPCR using more than 60,000 droplets. The sequencing protocol and accuracy have been verified to be reliable in our published articles^{1,2}.

Comment c: Digital droplet PCR (ddPCR) is a sensitive technique for identification of single variants. While it is appreciated that the authors validated some of the patients was both WES and ddPCR, there is no description of the sensitivity of the ddPCR assay for this site. The report a range of VAF% from 0.16 to 4%. A secondary validation with another assay such as amplicon sequencing which has far greater coverage than WES (often 50-100k reads per amplicon) would be important to confirm the findings.

Our response: We thank you for this insightful comment. ddPCR is indeed a highly sensitive method for detecting single nucleotide variants. All of the variants detected by WES were validated by ddPCR and extra 5 variants with a lower VAF. We tested the ddPCR system in negative controls and no-template controls in different ways, and the

probe always presented a clean background. These results show that the ddPCR assay for this site has good sensitivity and specificity.

In addition to validating our findings using WES and ddPCR, we agree with you that further validation using another high-sensitivity technique, such as amplicon sequencing, would strengthen our results. We performed amplicon sequencing on a subset of samples. To enhance detection sensitivity for low-frequency variants, we cloned the PCR products into plasmids. This approach allows the separation of individual DNA molecules, thus reducing the risk of the low-frequency variant being masked by the wild-type sequence. We then performed Sanger sequencing on multiple clones to detect the presence of the variant allele. This method allowed us to capture even low-frequency mutations at the target site. When there are more than 3 strains in the sample that are mutant by Sanger sequencing, it is considered a true positive. We verified the above results for two samples that showed positive results by WES and ddPCR, and the results were all true positives.

The combination of PCR amplification and clone sequencing increased the sensitivity of our detection. The results from Sanger sequencing corroborated the ddPCR findings, further confirming the presence of the low-frequency variants.

Figure legend: Validation of NAP1L1 p.D349E. PCR amplicons were cloned and subjected to Sanger sequencing analysis. The sequencing electrophoretograms are

shown (Left, wild-type allele; Right, mutant allele)

Comment d: Identification of the NAP1L1 p.D349E variant by FACS is reasonable, although there appears to be a considerable discrepancy between the reported VAF of the cTnT+ cells and the previously reported VAF in Supplemental Table S8. What is the unfractionated sample? This is not described in the text and accounts for approximately 50% of the cTnT+ sample. What are these cells? A more detailed description of this experiment and the relative contribution of NAP1L1 p.D349E positivity from each fraction should be included.

Our response: We appreciate your comments and recognize the need for clearer communication of the FACS results. The bar chart illustrates the VAF determined by ddPCR for different cell types sorted by FACS. Specifically: The *cTnT*-positive cells represent cardiomyocytes, and in patient P6, for example, the VAF of the NAP1L1 p.D349E variant was 8.8% in these cells.

The *cTnT*-negative cells represent non-cardiomyocytes, and in the same patient (P6), the VAF was much lower, at 0.13%. The unfractionated sample refers to the bulk myocardial before cell sorting, in which the VAF was determined using ddPCR on the whole tissue. This value reflects the average VAF across all cell types present in the tissue (including both cardiomyocytes and non-cardiomyocytes), and it matches the values reported in Supplementary Table S8. In this context, the unfractionated sample provides a baseline VAF for the entire tissue, which helps to compare the VAF specifically in cardiomyocytes versus non-cardiomyocytes.

We have now revised the manuscript to more clearly describe the experimental setup and the meaning of the unfractionated sample as a bulk measurement of the tissue before FACS sorting, along with the specific VAF values for each fraction (*cTnT*-positive and *cTnT*-negative) after sorting.

Figure legend: The fractional abundance of NAP1L1 p.D349E in cTnT+ and cTnT- cells from patients P6, R35, and R38, as well as in the unfractionated whole tissue sample from an adjacent tissue section.

Comment e: Minor point – The description of the isolation of DNA from the patients is limited. How much tissue was used, were multiple sites samples, was there any gross differences in fibrosis or other structural changes between study patients.

Our response: We appreciate your feedback regarding the DNA isolation process. In the revised manuscript, we have expanded the description to provide more detail.

For each patient, approximately 50 mg of myocardial tissue was used for DNA isolation. The tissue was collected from the left ventricular septum, as this region is commonly affected in HCM. In terms of structural differences, we did not observe significant gross fibrosis or other macrostructural changes during tissue collection between patients.

Comment f: Minor point – The labeling of the ddPCR graphs is very poor and confusing. Please include appropriate labels for positive signals and the source of each sample, this is especially true for the negative controls in Supplemental figure S2. It is entirely unclear the source of these samples.

Our response: We appreciate your feedback regarding the clarity of the ddPCR graphs and the need for better labeling. In response, we have revised the graphs to ensure clearer and more comprehensive labeling. Positive signals are now clearly indicated on each graph with appropriate color coding and annotations. For the negative controls in

Supplemental Figure S2, we have provided a clear description of the sample source, and the negative controls are now explicitly labeled to avoid confusion. These changes should improve the readability of the graphs and make it easier to distinguish between the various experimental conditions.

Figure legends: A, Representative somatic variants identified by ddPCR in the discovery cohort (D26). X axis, fluorescence intensity detected in the HEX-channel (channel 2, wild type); Y axis, fluorescence intensity detected in the FAM-channel (channel 1, mutant type); grey dots, droplets with background fluorescence; green dots, droplets with fluorescence detected in the HEX-channel; blue dots, droplets with fluorescence detected in the FAM-channel; orange dots, droplets with signals in both channels. B, Representative somatic variants identified by ddPCR in the discovery cohort (R7). C, Representative plot of ddPCR-negative NAP1L1 p.D349E (H5 and

H12). S2A, *Upper*, plot of ddPCR result on negative control H7; *Lower*, plot of ddPCR result on negative control H14; X axis, fluorescence intensity detected in the HEX-channel (channel 2, wild type); Y axis, fluorescence intensity detected in the FAM-channel (channel 1, mutant type); grey dots, droplets with background fluorescence; green dots, droplets with fluorescence detected in the HEX-channel.

2. Mouse model of NAP111 p.D349E

Comment a: Expression levels of AAV9-cTnT-hNAP111D349E are poorly characterized and do not match the clinical data. What is the transduction efficiency of the AAV9-cTnT-hNAP111D349E and AAV9-cTnT-hNAP111WT viruses?

Our response: We recognize the limitations of using animal models to fully replicate the chronic, low-frequency presence of NAP1L1 p.D349E observed in human myocardial tissue. In patients, the low-frequency presence of this variant is likely a result of a prolonged process involving chronic aseptic inflammation that stimulates surrounding cells over a period of years.

Animal models, particularly in the context of our study, may not accurately simulate this prolonged and gradual process. Instead, the use of high-dose viral infection in a short timeframe was intended to provide a more intense stimulus. This approach aims to capture the effects of NAP1L1 p.D349E in a more immediate and controlled manner over the lifespan of mice, compensating to some extent for the lack of prolonged chronic stimulation in the animal model.

The transduction efficiency of the AAV9-cTnT-hNAP111D349E and AAV9-cTnT-hNAP111WT viruses is shown in Figure A and B. Although this model does not perfectly mirror the extended duration of chronic inflammation in patients, it allows us to investigate the acute effects of NAP1L1 p.D349E on cardiomyocytes and observe its impact within a shorter experimental timeframe. We have clarified this point in the **Limitation** of revised manuscript to better explain the rationale behind our experimental design and its implications for interpreting the results:

Secondly, due to the challenge of replicating the impact of low-frequency somatic mutations in myocardial tissue over the course of several years-even decades-in the lifespan of mice, we introduced a model with high mutation load and overexpression.

This approach was necessary to recreate the phenotype of cardiac hypertrophy within a feasible experimental timeframe. However, it may not fully capture the slower, progressive nature of the disease observed in human patients.

present at later ages and with more severe phenotypes of hypertrophy, then a corresponding aged model should be performed.

In the human cohorts, were the affected patients hypertensive? What about patients with hypertension induced cardiac hypertrophy and/or heart failure, do they have an increased VAF % for the NAP111 p.D349E variant? Is possible that the NAP111 p.D349E variant predisposes patients to pressure-overload hypertrophy but not with the same mechanism as traditional sarcomeric forms of HCM? A discussion and explanation of this possibility along with the above data should be included.

Our response: We acknowledge your concern regarding the use of angiotensin II as a model. Our further study shows that NAP1L1 p.D349E over-expressing mice at 24 weeks of age, without angiotensin II treatment, do not exhibit significant differences in myocardial phenotype compared to NAP1L1 WT mice (Figure A to C). Echocardiography showed no difference in cardiac function (Figure D and E).

Given that myocardial cells in aging mice experience only approximately 24 months of pressure load, which does not fully replicate the long-term normal load experienced by patients, we used high-dose angiotensin II to simulate a prolonged normal pressure load through short-term overload. This approach provides a more intense stimulus, which helps to model the chronic stress that patients with NAP1L1 p.D349E might endure.

As we describe in the Methods section, in our human cohort, we specifically excluded patients with hypertension, valvular disease, congenital disease, infiltrative cardiomyopathy, or other diseases that could lead to abnormal loading conditions. We initially excluded secondary myocardial hypertrophy due to abnormal pressure loading. We hypothesize that the hypertrophy induced by NAP1L1 p.D349E is unrelated to traditional sarcomeric mutations. Our data suggest that patients with the NAP1L1 p.D349E mutation may have a subset of cardiomyocytes that cannot tolerate normal load even in the absence of hypertension.

Besides, we constructed Nap111 p.D349E knock in (KI) mice to investigate whether the pressure overload is necessary. The results suggested that even without the use of angiotensin II, KI mice at 12 weeks old also showed echocardiographic and pathological manifestations of cardiac hypertrophy, which confirmed that the

pathogenicity of this mutation is not dependent on pressure overload (Figure F to K).

As a clarification of this issue, we added in the Discussion as follows:

Our study demonstrated that the NAP1L1 p.D349E variant plays a significant role in the development of HCM. While angiotensin II infusion is commonly used to model hypertension-induced cardiac hypertrophy, it is important to recognize that this model does not perfectly replicate the mechanisms underlying genetic forms of HCM. In particular, sarcomeric mutations, which are central to most genetic forms of HCM, result in distinct cellular hypertrophy pathways³.

We hypothesize that the NAP1L1 p.D349E mutation contributes to a subset of myocardial cells that are less capable of tolerating normal physiological load. This hypothesis is supported by clinical observations that patients carrying the NAP1L1 p.D349E variant-despite lacking hypertension or other abnormal loading conditions-still present with hypertrophic changes. In this context, even normal pressure loads may lead to gradual hypertrophy over time due to an inherent susceptibility of the mutated cardiomyocytes.

However, due to the limitations of replicating the long-term effects of chronic normal load in animal models, we used angiotensin II to induce short-term pressure overload as a proxy for prolonged normal load. While this approach resulted in higher transduction efficiency and more pronounced hypertrophy in mice, it simulates the chronic stress that human cardiomyocytes with the NAP1L1 p.D349E variant may experience over years. This adaptation, though imperfect, allows us to better model the mutation's long-term effects.

Our findings suggest that the NAP1L1 p.D349E variant may predispose individuals to hypertrophy through a mechanism distinct from traditional sarcomeric mutations, potentially by altering cellular tolerance to normal physiological stress. This hypothesis aligns with our exclusion of patients with hypertension and other abnormal loading conditions, ensuring that our results reflect the intrinsic effects of the mutation rather than secondary causes of hypertrophy.

Figure legends: A, Schematic diagram depicting the experimental strategy used in

C57BL6/N mice. B, Representative image of heart and wheat germ agglutinin (WGA) staining and quantification of the WT and MT group. C, Representative hematoxylin and eosin staining Masson's trichrome staining and quantification showing the fibrotic area in WT and MT group. D, Representative M-mode echocardiograms of the left ventricle of the WT and MT group. E, Quantification of left ventricular end-diastolic anterior wall thickness (LVAW; d), left ventricular end-diastolic posterior wall thickness (LVPW; d), ejection fraction (EF), fractional shortening (FS). F and G, Schematic diagram depicting the experimental strategy used in C57BL6/N mice. H, Representative image of heart and WGA staining and quantification of the control and KI group. I, Representative hematoxylin and eosin staining Masson's trichrome staining and quantification showing the fibrotic area in control and KI group. J, Representative M-mode echocardiograms of the left ventricle of the WT and MT group. K, Quantification of left ventricular end-diastolic anterior wall thickness (LVAW; d), left ventricular end-diastolic posterior wall thickness (LVPW; d), ejection fraction (EF), fractional shortening (FS).

Comment c: The proposed model by the authors is that the presence of the NAP111 p.D349E variant even at very low VAF%'s (0.16%) induces cardiac hypertrophy through stimulation of the cGAS-STING-IFN signaling pathway leading to expression of cardiac hypertrophic-inducing cytokines. What is the minimal number of positive cardiomyocytes with the NAP111 p.D349E variant that produces the phenotype? The dose of AAV9 should be titrated down to determine the minimal VAF% that induces cardiac hypertrophy and dysfunction. Furthermore, the authors should perform in-vitro studies at low levels of transduction to validate their proposed mechanism. A trans-well assay with membrane separation of WT and mutant-transfected (MT) NRCMs at various MT transfection doses would also serve to support the hypothesis of a paracrine effect on WT cardiomyocytes induced by a small number of NAP111 p.D349E variant-containing CMs.

Our response: We appreciate your insightful suggestion regarding titrating AAV9 to determine the minimal VAF% that induces cardiac hypertrophy and dysfunction, and the recommendation of performing in-vitro studies at low levels of transduction to

validate the paracrine mechanism of the NAP1L1 p.D349E variant.

In response, we conducted the following experiments to further investigate the mechanism.

In vitro experiments:

a. Titration of mutant plasmid: We transfected NRCMs with varying doses of the mutant plasmid to assess the effect of different transfection efficiencies. Even at a transfection efficiency as low as 15.62%, cells transfected with the mutant NAP1L1 p.D349E plasmid still showed a significant increase in cell size compared to the WT group. This supports the hypothesis that even a small population of cardiomyocytes expressing the variant can induce a hypertrophic phenotype (Figure A to C).

b. Transwell assay: To explore the paracrine/autocrine mechanism, we performed a Transwell co-culture assay. NRCMs or cardiac fibroblasts were seeded in the lower chamber, and after 12 hours, the upper chamber was seeded with NRCMs transfected with either the mutant plasmid or siRNA. After 48 hours of co-culture, RNA was extracted from the lower chamber cells for qPCR analysis. We observed significant upregulation of ISG genes in both cardiomyocytes and fibroblasts co-cultured with mutant-transfected or siRNA-treated NRCMs, indicating that paracrine signaling from mutant cells can influence neighboring wild-type cells (Figure D to G).

These results support our proposed model that a small number of cells carrying the NAP1L1 p.D349E variant can stimulate the cGAS-STING-IFN signaling pathway, promoting a hypertrophic response in neighboring wild-type cells via a paracrine mechanism. We also added these results in the revised manuscript as follows:

To verify the simulation of paracrine/autocrine, we performed a trans-well assay with membrane separation of control and *siNap111* transfected NRCMs. The results supported the hypothesis of a paracrine effect on WT cardiomyocytes induced by a small number of NAP111 p.D349E variant-containing CMs (Figure 2K and Supplementary Figure S5H).

For validation, we also performed a trans-well assay with membrane separation of WT and mutant-transfected NRCMs. The results of qPCR supported that somatic mutation *Nap111* p.D349E trigger IFN response of neighbor cells (Figure 4P). We speculated that

the acquisition of NAP1L1 p.D349E in some cardiomyocytes would lead to the formation of a local microenvironment of immune activation, affecting nearby cardiomyocytes and cardiac fibroblasts.

In vivo considerations: We did not conduct in vivo experiments to determine the minimal number of NAP1L1 p.D349E-positive cardiomyocytes required to produce the phenotype. While this is a valuable suggestion, we believe that VAF is not the only variable that has been implicated in the pathogenic process of the mutation, but also includes duration of stimulation, exposure to reactive oxygen species, secondary hits, etc. The duration of exposure to the mutation may be a more critical variable, as long-term chronic low-frequency stimulation may play an important role. Unfortunately, it is challenging to replicate such low-frequency, long-term effects in an in vivo experimental setting. As can be expected, the minimum number of NAP1L1 p.D349E positive cardiomyocytes required to produce the phenotype is affected by the duration of overexpression and the amount of angiotensin II used in the model construction. Due to these limitations, the minimum number of NAP1L1 p.D349E positive cardiomyocytes required to study the phenotype has limited our understanding of the role of low-frequency somatic mutations in patient cardiomyocytes.

Figure legends: A to C Effect and quantification of transfection with different amounts of DNA on cardiomyocyte hypertrophy; D to G, A trans-well assay using membrane separation revealed upregulation of ISG genes in mutant-transfected or siRNA-treated NRCM co-cultured cardiomyocytes and fibroblasts.

Comment 3: Rescue of hypertrophy with inhibition of cGAS-STING pathway – The authors demonstrate that with inhibition of the cGAS-STING using both a small-molecule inhibitor and antibody, there is prevention of cardiac hypertrophy in animals over-expressing the NAP1L1 p.D349E variant in the context of angiotensin II treatment (pressure overload). While these data are compelling, there is no corresponding controls for C-176 and anti-mouse IFNAR1 antibodies. What is the effect on WT hearts at baseline or exposed to angiotensin II? These data would be critical prior to any attempt of clinical translation for this work.

Our response: We thank you for raising this important point. We have conducted additional experiments to assess the effects of C-176 and anti-mouse IFNAR1 antibodies on wild-type (WT) hearts at baseline and treated with angiotensin II. In unaffected cardiomyocytes, the cGAS-STING pathway is naturally suppressed⁴. Consequently, there was no significant benefit observed from C-176 or anti-IFNAR1 antibodies at baseline mice or mice treated with angiotensin II. No abnormality on echocardiography or cardiomyocyte abnormalities, including hypertrophy, were observed (Figure A to G).

Revised Discussion:

Our study demonstrates the protective effect of cGAS-STING pathway inhibition on hypertrophy in NAP1L1 p.D349E-overexpressing mice subjected to angiotensin II-induced pressure overload. To address the potential impact of these inhibitors on WT hearts, we conducted additional experiments using C-176 and anti-IFNAR1 antibodies at baseline mice or mice treated with angiotensin II.

In unaffected cardiomyocytes, the cGAS-STING pathway is naturally suppressed, and we observed no significant benefit from pathway inhibition at baseline mice⁴. No abnormality on echocardiography or cardiomyocyte abnormalities, including hypertrophy, were observed. These findings provide further support for the cGAS-STING pathway as a therapeutic target in hypertrophic cardiomyopathy.

Figure legends: A, Schematic diagram depicting the experimental strategy used in C57BL6/N mice. B and C, Representative image of heart and wheat germ agglutinin (WGA) staining and quantification of the WT mice treated with C-176 and anti-mouse IFNAR1 antibodies at baseline or exposed to angiotensin II. D and E, Representative

hematoxylin and eosin staining Masson's trichrome staining and quantification showing the fibrotic area of the WT mice treated with C-176 and anti-mouse IFNAR1 antibodies at baseline or exposed to angiotensin II. F, Representative M-mode echocardiograms of the left ventricle of the WT mice treated with C-176 and anti-mouse IFNAR1 antibodies at baseline or exposed to angiotensin II. G, Quantification of left ventricular end-diastolic anterior wall thickness (LVAW; d), left ventricular end-diastolic posterior wall thickness (LVPW; d), ejection fraction (EF), fractional shortening (FS).

Comment: Minor points: There are multiple grammatical and spelling errors throughout the manuscript.

Our response: Thank you for pointing out the grammatical and spelling errors in the manuscript. We have carefully reviewed the entire manuscript and corrected any grammatical and typographical issues. We appreciate your attention to detail and have made the necessary revisions to ensure the manuscript is clear and accurate.

Reviewer #2

Comment 1: A recent study reported that NAP1L1 (PMID: 37593048) is a regulator of cardiac fibrosis and is upregulated in ischemic cardiomyopathy patient hearts and regulates cardiac fibrosis by inhibiting YAP1 ubiquitination and degradation. The authors should comment on this how this Nap1l1 variant may function in the cardiac fibroblasts and possibly contribute to HCM observed.

Our response: We appreciate you bringing attention to the recent study (PMID: 37593048) that highlights the role of NAP1L1 as a regulator of cardiac fibrosis in ischemic cardiomyopathy. While the study emphasizes NAP1L1's role in cardiac fibroblasts and its regulation of YAP1 ubiquitination and degradation, our study focuses on the NAP1L1 p.D349E variant and its effects on cardiomyocytes in HCM. Nevertheless, the findings on NAP1L1's role in fibroblasts raise intriguing possibilities regarding its function in other cardiac cell types, including its potential involvement in fibrosis in HCM.

Revised Discussion:

Shan *et al* reported NAP1L1 as a regulator of cardiac fibrosis, particularly in ischemic cardiomyopathy, through inhibition of YAP1 ubiquitination and degradation in cardiac fibroblasts⁵. This raises the possibility that the NAP1L1 p.D349E variant observed in our HCM cohort may also exert effects in cardiac fibroblasts, contributing to the development of fibrosis and the hypertrophic phenotype. While our study primarily focuses on cardiomyocytes, further investigation into the role of this variant in fibroblast biology could reveal additional mechanisms by which NAP1L1 p.D349E contributes to HCM pathogenesis.

Given that fibrosis is a common feature in both ischemic and hypertrophic cardiomyopathy, the interaction of NAP1L1 with pathways regulating cardiac remodeling, such as YAP1, may be relevant. Future studies exploring the role of NAP1L1 in cardiac fibroblasts, as well as the crosstalk between cardiomyocytes and fibroblasts, could provide valuable insights into the broader impact of the NAP1L1 p.D349E variant in HCM and other forms of cardiomyopathy.

Comment 2: Similar to the first comment, it appears that cGAS interaction and

expression levels were affected by D349E (fig. 3L, 4A, and 5G). Can the authors confirm whether the downstream activation of IFN signaling is due to altered cGAS stability or enhanced dsDNA release and sensing?

Our response: Thank you for your insightful question regarding the mechanism underlying the activation of IFN signaling in the context of the NAP1L1 p.D349E variant. Based on our data and further experiments, we have investigated whether the downstream activation of IFN signaling is primarily due to altered cGAS stability or enhanced dsDNA release and sensing.

In Figures 3L, 4A, and 5G, we observed that cGAS interacts with nucleosomes in the nucleus, and the NAP1L1 p.D349E variant disrupts this interaction, leading to cGAS release into the cytoplasm. To discern the primary factor driving downstream activation of the cGAS-STING pathway, we conducted additional experiments.

We performed a Western blot analysis after transfecting cells with plasmids or siRNA and treating them with cycloheximide, a protein synthesis inhibitor. After a 24-hour transfection period to allow for protein translation, and an additional 24-hour treatment with CHX, we found that Cgas levels remained unchanged between the groups. However, p-Sting levels were elevated, indicating that downstream signaling was activated even when cGAS levels did not differ significantly (Figure A to D).

These results suggest that the activation of downstream signaling is not solely dependent on the levels of cGAS but may also be influenced by other factors, such as the stability of cGAS or the availability of dsDNA. The increase in cGAS observed in Figure 3L might reflect changes in its cellular distribution or stability, which could contribute to the observed signaling activation.

New results

Figure legends: A and B, Western blot images and quantification of Cgas and p-Sting level under protein synthesis inhibitor cycloheximide after transfecting cells with WT or Nap111 p.D349E plasmids. C and D, Western blot images and quantification of Cgas and p-Sting level under protein synthesis inhibitor cycloheximide after transfecting

cells with control siRNA or Nap111 siRNA.

Comment 3: Data presented in fig. 4A alone did not support the claim that Nap111 variants affect nucleosome assembly. Did the authors find any DNA damage-related gene sets in the RNA-seq presented in fig. S5A? In addition, the authors should show the effect of the D349E variant on Nap111 expression and nucleosome assembly.

Our response: Thank you for highlighting the need for additional data to support the claim that NAP1L1 variants affect nucleosome assembly. We have conducted further experiments to address this concern.

DNA Damage-Related Gene Sets: In addition, we analyzed RNA-seq data presented in Figure S5A for the presence of DNA damage-related gene sets. Our analysis revealed that cells expressing the D349E variant exhibited upregulation of DNA damage response genes, which aligns with the hypothesis that the variant may impact nucleosome stability and contribute to DNA damage (A). We added in the manuscript as follows: DNA damage response related genes were upregulated in the *Nap111* knockdown group (Supplementary Figure S5A).

Impact on NAP1L1 Expression: We performed a series of experiments to evaluate the effect of the D349E variant on NAP1L1 expression. Cells were transfected with empty vector, wild-type NAP1L1 plasmid, and mutant NAP1L1 p.D349E plasmid. Western blot analysis revealed that NAP1L1 protein levels were significantly upregulated in cells transfected with the wild-type plasmid compared to the empty vector group. In contrast, cells transfected with the mutant D349E plasmid showed a significant downregulation of NAP1L1 protein. This downregulation is likely due to resource competition during protein synthesis and the rapid degradation of misfolded mutant proteins, which may contribute to similar phenotypes observed in cells with NAP1L1 knockdown (Figure B and C).

Nucleosome Assembly: To investigate the impact of the D349E variant on nucleosome assembly, we conducted experiments in NRCMs. Cells were transfected with plasmids or siRNA, and samples were harvested 48 hours post-transfection. Micrococcal Nuclease was used to cleave DNA not wrapped around nucleosomes. The DNA that was wrapped around nucleosomes was then purified and analyzed by agarose gel

electrophoresis. The results revealed that the DNA content in cells transfected with the mutant D349E plasmid or with NAP1L1 knockdown was significantly lower compared to the control group, indicating a reduced number of nucleosomes. This suggests that the process of assembling complete nucleosomes is disrupted in these cells, further supporting our hypothesis that the NAP1L1 variant impairs nucleosome stability and assembly (Figure D to G).

We believe these additional data support our claim and provide a clearer understanding of how the NAP1L1 p.D349E variant affects nucleosome assembly and expression.

Figure legends: A, Heatmap of upregulated interferon-stimulated genes and DNA damage response genes. B and C, Western blot image and quantification of Nap111 in the vector, WT and MT group. D to G, Agarose gel electrophoresis showing the nucleosome-bound DNA content, representing the number of nucleosomes in the WT and MT group.

Comment 4: *The authors should clearly mention in the legends of fig. 4A which antibody was used for the pull-down and include the isotype controls in the experiment.*

Our response: We appreciate your suggestion regarding clarification of the antibodies used in the pull-down experiment shown in **Figure 4A**. To address this:

The antibodies used for the pull-down were anti-HA for HA-tagged NAP1L1 detection, anti-Flag for the FLAG-tagged proteins, and anti-IgG as the isotype control.

The isotype control, IgG, was included in the experiment to validate the specificity of the pull-down assays.

We have revised the figure legend to include this information and to clearly mention the use of isotype controls, as requested.

Figure legend: Co-IP of the Nap111 p.D349E or wild type with H2a, H2a.x, H2a.z and H2b; The antibodies used for pulling were IgG, HA, and Flag, and IgG is the isotype control group.

Comment 5: The results shown in fig. 4G are not enough to support that dsDNA released to the cytosol is not from mitochondria. The authors should stain cells as shown in fig. 4F with mitochondrial markers to exclude mitochondrial damage and mtDNA release (i.e. –analysis of TFAM outside of mitochondrial networks).

Our response: We appreciate your suggestion to investigate whether dsDNA release to the cytosol could originate from mitochondrial damage. To address this, we performed additional experiments:

After transfecting cells with wild-type and mutant plasmids, we separated the cytosolic and mitochondrial fractions and measured Tfam protein levels in the cytosolic fraction. No significant differences in Tfam levels were observed between the two groups, suggesting that mtDNA release is not a major contributor to the cytosolic dsDNA detected. Furthermore, we conducted fluorescence staining for both mitochondria and Tfam in cells transfected with wild-type and mutant plasmids. Mitochondria maintained a largely tubular structure in both groups, and most Tfam remained within the mitochondrial network. While a small fraction of Tfam was detected outside of the mitochondria, this was minimal and showed no significant difference between the two groups.

These findings suggest that the dsDNA observed in the cytosol is not primarily due to mitochondrial damage or mtDNA release.

Figure legends: A and B, Western blot image and quantification of Tfam in the cytosol and mitochondria; C, Mito., Tfam and DAPI staining in WT and MT groups

Comment 6: The authors have shown that mRNA levels of cytokines are increased after Nap111 silencing or D349E expression (fig. 2I, 4L and 4N). cGAS is not a robust inducer of pro-inflammatory cytokines. Thus, this data could be strengthened if the authors can show the levels of released cytokines in culture media supernatants or tissue extracts.

Our response: We appreciate your suggestion on released cytokines. To address the reviewer's concern, we have performed additional experiments to measure cytokine release in the supernatant following NAP1L1 knockdown or MT expression. Cells were transfected with either wild-type or mutant NAP1L1 plasmids, or with Nap111 siRNA and Control siRNA, and the levels of Ifna and Ifnb in the culture media were measured after 48 hours. The results show that the levels of both Ifna and Ifnb were significantly elevated in the supernatant following expression of the mutant NAP1L1 or Nap111 silencing, supporting the notion that NAP1L1 p.D349E or knockdown of NAP1L1 promotes cytokine release. This strengthens the hypothesis that the cGAS-STING pathway is involved in the observed inflammatory response.

Figure legends: A to D, the level of Ifna and Ifnb in culture media supernatant from cell culture treated with wild-type or mutant NAP1L1 plasmids, or with Nap111 siRNA and Control siRNA. E and F, the level of Ifna and Ifnb in tissue extract from the cardiac tissue of WT and MT treated mice.

Comment 7: *For mouse experiments, the authors express human Nap111 variants in the presence of WT mouse Nap111. To better understand the function and effect, the authors should use genetically engineered models in their study. Can the authors knockdown/knockout endogenous Nap111 then express the D349E variant to cleanly assess the role of this protein in the absence of WT protein.*

Our response: We appreciate your suggestion to use genetically engineered models for a cleaner assessment of the role of the NAP1L1 p.D349E variant in the absence of wild-type NAP1L1. In fact, we initially explored the outcome of a cardiomyocyte-specific Nap111 p.D349E genetically engineered mouse model. The hypertrophy-associated cardiac phenotype observed in these mice was consistent with the results obtained from adeno-associated-virus-mediated overexpression of the NAP1L1 p.D349E variant.

However, we ultimately decided not to pursue the genetically engineered mouse model in our final experiments. Since all cardiomyocytes in these mice would carry the mutation, we felt that this approach might artificially exacerbate the phenotype caused by somatic mutations, potentially deviating from the more nuanced effects observed in human cases where the mutation is present in only a subset of cells. Therefore, we chose adeno-associated-virus-mediated overexpression of the NAP1L1 p.D349E variant as a model that more closely mimics the somatic mutation state in humans.

We believe this approach allows us to better represent the sporadic nature of somatic mutations in human cardiomyocytes, while still providing valuable insights into the functional impact of the NAP1L1 p.D349E variant.

Figure legend: A and B, Schematic diagram depicting the experimental strategy used in C57BL6/N mice. C, Representative image of heart and WGA staining and quantification of the control and KI group. D, Representative hematoxylin and eosin

staining Masson's trichrome staining and quantification showing the fibrotic area in control and KI group. E, Representative M-mode echocardiograms of the left ventricle of the WT and MT group. F, Quantification of left ventricular end-diastolic anterior wall thickness (LVAW; d), left ventricular end-diastolic posterior wall thickness (LVPW; d), ejection fraction (EF), fractional shortening (FS).

Comment 8: The authors should quantify the IF showed in fig 2A, 2B, 4E, 4F and S7B.

Our response: Thank you for your suggestion. We have conducted additional analyses to quantify the fluorescence intensities in these images. The updated quantification data will be included in the revised manuscript to provide a more comprehensive view of the expression levels and localization of the relevant proteins. We appreciate your input and believe that these additions will enhance the clarity of our results.

Figure legends: IF and quantification of figure 2A, 2B, 2G, 4E, 4F and S7B.

Reviewer #3

Comment: Major

Introduction:

The contents of the introduction do not effectively convey the hypothesis to the reader. It would be improved with the addition of a hypothesis describing the putative mechanism on HCM (page 4 lines 15-17)

Our response: Thank you for your valuable suggestion. We agree that clearly stating the hypothesis will improve the clarity of the introduction. We have revised the introduction to explicitly include our hypothesis regarding the putative mechanism of HCM associated with the NAP1L1 p.D349E variant.

Revised Introduction:

In recent studies, non-classical genetic alterations have been linked to various cardiovascular diseases^{6,7}. Through somatic mutation screening in two sporadic HCM cohorts, we addressed the NAP1L1 p.D349E mutation to be a potential contributor to HCM. We propose that the p.D349E mutation destabilizes nucleosome formation, leading to aberrant release of DNA into the cytosol. This release triggers the cGAS-STING pathway, which results in the production of pro-inflammatory cytokines, leading to cardiac hypertrophy by creating a hyper-immune microenvironment. By exploring this mechanism, we aim to uncover new insights into the molecular basis of HCM and identify potential therapeutic targets.

Comment: I am concerned that the overall approach of using ddPCR as well as the next generation sequencing techniques are not robust enough to evaluate a mutation with a frequency this low. Sanger sequencing similarly would not be an effective means for determining the absence or presence of this variant in the heart (or other) tissue. As it stands, it is not able to be determined if the variant of interest is bona-fide or artifact of the chosen assays. This is true for both the tissue and blood based approaches. Something like FISH (or mutaFISH) confirmation and/or UMI-based, deep error-corrected NGS would be necessary to confirm the presence of the variant of interest in the tissue given the low VAF in the tissue.

Our response: We appreciate your concern regarding the robustness of our mutation detection methods. To validate the authenticity of the mutations, we employed amplicon sequencing, as recommended by Reviewer #1. Amplicon sequencing is a targeted approach that amplifies specific regions of interest in the genome using PCR before sequencing, providing high coverage and sensitivity for detecting low-frequency variants. This method allowed us to confirm the presence of the variant with greater accuracy compared to other techniques.

To enhance detection sensitivity for low-frequency variants, we cloned the PCR products into plasmids. This approach allows the separation of individual DNA molecules, thus reducing the risk of the low-frequency variant being masked by the wild-type sequence. We then performed Sanger sequencing on multiple clones to detect the presence of the variant allele. This method allowed us to capture even low-frequency mutations at the target site. When there are more than 3 strains in the sample that are mutant by Sanger sequencing, it is considered a true positive. We verified the above results for two samples that showed positive results by WES and ddPCR, and the results were all true positives.

The combination of PCR amplification and clone sequencing increased the sensitivity of our detection. The results from Sanger sequencing corroborated the ddPCR findings, further confirming the presence of the low-frequency variants.

Figure legend: Validation of NAP1L1 p.D349E. PCR amplicons were cloned and subjected to Sanger sequencing analysis. The sequencing electrophoretograms are shown (Left, wild-type allele; Right, mutant allele).

Comment: The animal models do seem to exert an effect. However, this model does not seem to accurately recapitulate the circumstances in the human physiology identified in the two cohorts. By looking at the flag staining (SF S6), it would appear that far more than 4-8% of cells are affected, making it difficult to triangulate the magnitude of the effect for a much smaller proportion of cells.

Our response: We recognize the limitations of using animal models to fully replicate the chronic, low-frequency presence of NAP1L1 p.D349E observed in human myocardial tissue. In patients, the low-frequency presence of this variant is likely a result of a prolonged process involving chronic aseptic inflammation that stimulates surrounding cells over a period of years.

Animal models, particularly in the context of our study, may not accurately simulate this prolonged and gradual process. Instead, the use of high-dose viral infection in a short timeframe was intended to provide a more intense stimulus. This approach aims to capture the effects of NAP1L1 p.D349E in a more immediate and controlled manner over the lifespan of mice, compensating to some extent for the lack of prolonged chronic stimulation in the animal model.

Although this model does not perfectly mirror the extended duration of chronic inflammation in patients, it allows us to investigate the acute effects of NAP1L1 p.D349E on cardiomyocytes and observe its impact within a shorter experimental timeframe. We have clarified this point in the revised manuscript to better explain the rationale behind our experimental design and its implications for interpreting the results.

Comment: Are the authors able to further evaluate any other heart region or non-dividing tissue in their cohorts for the SNP using ddPCR to determine cardiac specificity and or intra-cardiac regionality?

Our response: We appreciate your suggestion to further evaluate other heart regions or non-dividing tissues using ddPCR to determine cardiac specificity and intra-cardiac regionality of the SNP. Unfortunately, due to the limited availability of tissue samples,

we are unable to conduct additional analyses in other heart regions or non-dividing tissues in our current cohort. The heart tissue available for our study was already fully utilized for the experiments reported. Furthermore, access to additional cardiac tissue or non-dividing tissues from our cohort is currently restricted by ethical constraints.

However, in our somatic calling procedure, we included peripheral blood as a control in our somatic mutation calling process. This allowed us to rule out the presence of the NAP1L1 p.D349E variant in non-cardiac cells, thereby supporting the specificity of the mutation to cardiac tissues in the current analysis. While this does not directly evaluate regional variation within the heart, it provides confidence that the mutation is not widespread in other cell types.

We recognize the value of additional analyses for determining intra-cardiac regionality, and future studies with expanded sample availability will be necessary to explore both cardiac specificity and regional variation of the NAP1L1 p.D349E variant. We acknowledge the importance of this point and will aim to incorporate these analyses in future investigations, particularly in studies involving larger cohorts with more comprehensive tissue sampling.

Comment: Have the investigators considered the effects of polyploidy in the setting of cardiac tissue? This would potentially confound the reported VAFs to be artificially higher than they actually are.

Our response: We appreciate your insightful comment regarding the potential effects of polyploidy in cardiac tissue. We acknowledge that we did not initially consider the effects of polyploidy when analyzing the VAFs. Polyploidy, common in cardiomyocytes, could indeed lead to a dilution of the VAF, making the observed VAF appear lower than it might be in a fully diploid context.

However, given the characteristics of multinucleated cells, particularly in the case of the NAP1L1 p.D349E mutation, we believe that the potential dilution of the VAF due to polyploidy is unlikely to significantly affect our hypothesis. If multiple nuclei within a polyploid cell harbor the mutation, the cell may produce higher amounts of dsDNA, leading to enhanced activation of the cGAS-STING pathway, which could amplify the paracrine effect. Therefore, even with a lower apparent VAF due to polyploidy, the

functional impact of the mutation on neighboring cells and the overall tissue environment could still be significant.

We have included the following in our discussion:

Polyploidy may dilute the observed VAF, making it appear lower than in a diploid environment. This dilution effect is particularly important in cardiomyocytes, where polyploidy is common and may affect our measurements of mutation frequency to some extent. However, even with this dilution, multinucleated cells containing the NAP1L1 p.D349E mutation may produce large amounts of double-stranded DNA (dsDNA), leading to higher activation of the cGAS-STING pathway and amplifying paracrine effects. Therefore, although VAF may be lower, the functional impact of the mutation on the tissue microenvironment may still be large.

Comment: It is notable that no phenotypic changes are present in animal hearts without the addition of Ang-II. Do the authors have any insight as to why this is the case?

Our response: We acknowledge your concern regarding the use of angiotensin II as a model. Our further study shows that NAP1L1 p.D349E over-expressing mice at 24 weeks of age, without angiotensin II treatment, do not exhibit significant differences in myocardial phenotype compared to NAP1L1 WT mice.

Given that myocardial cells in aging mice experience only approximately 24 months of pressure load, which does not fully replicate the long-term normal load experienced by patients, we used high-dose angiotensin II to simulate a prolonged normal pressure load through short-term overload. This approach provides a more intense stimulus, which helps to model the chronic stress that patients with NAP1L1 p.D349E might endure.

As we describe in the Methods section, in our human cohort, we specifically excluded patients with hypertension, valvular disease, congenital disease, infiltrative cardiomyopathy, or other diseases that could lead to abnormal loading conditions. We initially excluded secondary myocardial hypertrophy due to abnormal pressure loading. We hypothesize that the hypertrophy induced by NAP1L1 p.D349E is unrelated to traditional sarcomeric mutations. Our data suggest that patients with the NAP1L1 p.D349E mutation may have a subset of cardiomyocytes that cannot tolerate normal

load even in the absence of hypertension.

Besides, we constructed Nap111 p.D349E knock in (KI) mice to investigate whether the pressure overload is necessary. The results suggested that even without the use of angiotensin II, KI mice at 12 weeks old also showed echocardiographic and pathological manifestations of cardiac hypertrophy, which confirmed that the pathogenicity of this mutation is not dependent on pressure overload.

As a clarification of this issue, we added in the Discussion as follows:

Our study demonstrated that the NAP1L1 p.D349E variant plays a significant role in the development of HCM. While angiotensin II infusion is commonly used to model hypertension-induced cardiac hypertrophy, it is important to recognize that this model does not perfectly replicate the mechanisms underlying genetic forms of HCM. In particular, sarcomeric mutations, which are central to most genetic forms of HCM, result in distinct cellular hypertrophy pathways³.

We hypothesize that the NAP1L1 p.D349E mutation contributes to a subset of myocardial cells that are less capable of tolerating normal physiological load. This hypothesis is supported by clinical observations that patients carrying the NAP1L1 p.D349E variant-despite lacking hypertension or other abnormal loading conditions-still present with hypertrophic changes. In this context, even normal pressure loads may lead to gradual hypertrophy over time due to an inherent susceptibility of the mutated cardiomyocytes.

However, due to the limitations of replicating the long-term effects of chronic normal load in animal models, we used angiotensin II to induce short-term pressure overload as a proxy for prolonged normal load. While this approach resulted in higher transduction efficiency and more pronounced hypertrophy in mice, it simulates the chronic stress that human cardiomyocytes with the NAP1L1 p.D349E variant may experience over years. This adaptation, though imperfect, allows us to better model the mutation's long-term effects.

Our findings suggest that the NAP1L1 p.D349E variant may predispose individuals to hypertrophy through a mechanism distinct from traditional sarcomeric mutations, potentially by altering cellular tolerance to normal physiological stress. This hypothesis

aligns with our exclusion of patients with hypertension and other abnormal loading conditions, ensuring that our results reflect the intrinsic effects of the mutation rather than secondary causes of hypertrophy.

Figure legend: A and B, Schematic diagram depicting the experimental strategy used in C57BL6/N mice. C, Representative image of heart and WGA staining and quantification of the control and KI group. D, Representative hematoxylin and eosin staining Masson's trichrome staining and quantification showing the fibrotic area in control and KI group. E, Representative M-mode echocardiograms of the left ventricle of the WT and MT group. F, Quantification of left ventricular end-diastolic anterior wall thickness (LVAW; d), left ventricular end-diastolic posterior wall thickness (LVPW; d), ejection fraction (EF), fractional shortening (FS).

Comment: Minor

Nomenclature of genes and gene products – Genes in Italics and gene products in normal type. Is “NAP1-like” the gene product of NAP1L1?

Our response: Thank you for your comments on the nomenclature of gene names and gene products. We apologize for any confusion caused by the use of nomenclature in the manuscript. To clarify, according to standard convention, genes should be italicized, while gene products should be in normal font. In our study, "NAP1-like" refers to protein products of the NAP1L family. Mammals possess five nucleosome assembly protein 1-like (NAP1L) proteins, as follows: NAP1L1, NAP1L2, NAP1L3, NAP1L4 and NAP1L5.

We have revised the manuscript to ensure that this nomenclature is applied consistently and that gene names and their products are clearly distinguished according to convention. We appreciate your attention to this detail and apologize for any confusion that may have been caused.

Comment: No reference given for what genome build was used in alignment of the sequencing. In GRCh38, the genomic position of the mutation in question is Chr12:76050543 whereas the location shown in the BAM chr12:76444323

Our response: We apologize for the misunderstanding caused by the undefined reference genome version. We clarified in the revised manuscript that we use hg19 as the reference genome.

Comment: The SNV in question has been identified in COSMIC:
<https://cancer.sanger.ac.uk/cosmic/mutation/overview?id=177063780>

Our response: We appreciate you pointing out that this variant has been identified in the COSMIC database. We acknowledge the importance of this finding and incorporate this information into our manuscript to provide a more comprehensive context for the relevance of the mutation. We added in the **Discussion** of the revised manuscript as follows:

NAP1L1 p.D349E has been identified in COSMIC with Genomic Mutation ID COSV99673941, further implicating that this mutation could be involved in pathogenicity as a somatic mutation.

Comment: Manual review of IGV was outlined in the methods but the criteria undertaken to further filter the variants were not defined.

Our response: We apologize for the lack of clarity in this section. During the manual review of variants in IGV, we used the following criteria to further filter variants:

Read Depth: Variants were included only if they had a minimum read depth of 20 reads to ensure sufficient coverage.

Strand Bias: We excluded variants that demonstrated significant strand bias, which we defined as a ratio of variant-supporting reads on one strand being more than 70% compared to the other.

Base Quality: Variants were reviewed for base quality, and those with an average Phred score of <20 were excluded from further analysis.

Mapping Quality: Only variants with a mapping quality of >30 were considered reliable and included in the final analysis.

Visual Confirmation: We manually inspected the variant positions in IGV to confirm the presence of the variant across multiple reads, ensuring consistent alignment and avoiding artifacts from sequencing or mapping errors.

These criteria were applied uniformly across all samples to ensure consistency and reliability of the variants selected for further validation.

Comment: Can density of total and small arteriolar area be quantified? (Supplementary Figure S6D)

Our response: Thank you for your feedback regarding the quantification of the density of total and small arteriolar areas shown in Supplementary Figure S6D. We have now performed a quantitative analysis to provide more detailed insights.

We quantified the density of both total and small arteriolar areas using the following methods:

Image Analysis: We utilized image analysis software to measure the total and small arteriolar areas from the stained tissue sections. The software allowed for precise delineation and measurement of the arteriolar regions.

Quantification: The density of total and small arteriolar areas was calculated by dividing the area occupied by the arterioles by the total area of the tissue section. This was expressed as a percentage to facilitate comparison.

Statistical Analysis: Statistical tests were performed to compare the density between experimental groups and controls. the significance was assessed using appropriate statistical tests.

We have updated Supplementary Figure S6D to include the quantification data and have provided a detailed description of the quantification methods in the revised methods section of the manuscript.

Figure legend: F4/80 staining showing the macrophage infiltration and quantification of the density of total and small arteriolar area.

Comment: Page 1, Line 6 – should read “somatic variants are”

Page 1, Line 13 – “contributes”

Our response: Thank you for pointing out the grammatical and spelling errors in the manuscript. We have carefully reviewed the entire manuscript and corrected any grammatical and typographical issues. We appreciate your attention to detail and have made the necessary revisions to ensure the manuscript is clear and accurate.

Reference

- 1 Hong, T. *et al.* Somatic MAP3K3 and PIK3CA mutations in sporadic cerebral and spinal cord cavernous malformations. *Brain* **144**, 2648–2658, doi:10.1093/brain/awab117 (2021).
- 2 Hong, T. *et al.* High prevalence of KRAS/BRAF somatic mutations in brain and spinal cord arteriovenous malformations. *Brain* **142**, 23–34, doi:10.1093/brain/awy307 (2019).
- 3 Schmitt, J. P. *et al.* Consequences of pressure overload on sarcomere protein mutation-induced hypertrophic cardiomyopathy. *Circulation* **108**, 1133–1138, doi:10.1161/01.CIR.0000086469.85750.48 (2003).
- 4 Pathare, G. R. *et al.* Structural mechanism of cGAS inhibition by the nucleosome. *Nature* **587**, 668–672, doi:10.1038/s41586-020-2750-6 (2020).
- 5 Li, T. *et al.* Nucleosome assembly protein 1 like 1 (NAP1L1) promotes cardiac fibrosis by inhibiting YAP1 ubiquitination and degradation. *MedComm (2020)* **4**, e348, doi:10.1002/mco2.348 (2023).
- 6 Mustjoki, S. & Young, N. S. Somatic Mutations in “Benign” Disease. *N Engl J Med* **384**, 2039–2052, doi:10.1056/NEJMra2101920 (2021).
- 7 Heimlich, J. B. & Bick, A. G. Somatic Mutations in Cardiovascular Disease. *Circ Res* **130**, 149–161, doi:10.1161/CIRCRESAHA.121.319809 (2022).

Full title: Somatic NAP1L1 p.D349E promotes cardiac hypertrophy through cGAS-STING-IFN signaling

Running title: NAP1L1 p.D349E promotes cardiac hypertrophy

Authors' names and affiliations: Cheng Lv^{1,5}, Xiayidan Alimu^{1,5}, Xiao Xiao^{1,5}, Fei Wang^{1,5}, Jizheng Wang^{1,5}, Shuiyun Wang², Guixin Wu¹, Yu Zhang¹, Yue Wu³, Houzao Chen⁴, Rutai Hui¹, Lei Song^{1,6}, Yibo Wang^{1,6,7}

¹State Key Laboratory of Cardiovascular Disease, Fuwai Hospital, National Center for Cardiovascular Diseases, Chinese Academy of Medical Sciences and Peking Union Medical College, Beijing, China.

²Department of Cardiac Surgery, Fuwai Hospital, Chinese Academy of Medical Sciences and Peking Union Medical College, Beijing, China.

³Department of Cardiovascular Medicine, The First Affiliated Hospital of Xi'an Jiaotong University, Xi'an, China.

⁴State Key Laboratory of Common Mechanism Research for Major Diseases, Department of Biochemistry and Molecular Biology, Institute of Basic Medical Sciences, Chinese Academy of Medical Sciences and Peking Union Medical College, Beijing, China.

⁵The first five authors contributed equally to this article.

⁶Correspondence: songlqd@126.com and yibowang@hotmail.com

⁷Lead contact, Yibo Wang, Ph.D., Fuwai Hospital, National Center for Cardiovascular Diseases, Chinese Academy of Medical Sciences and Peking Union Medical College,

167 Beilishi Rd, Beijing, 100037, China. E-mail: yibowang@hotmail.com, Phone
number: +86-10-60866112

Reviewer #1

Comment: However, the sensitivity of the primary method to detect the somatic variant, namely ddPCR, still is not adequately addressed. The authors used the traditional method of cloning the region by PCR and then performing Sanger sequencing to identify the variant. While this method demonstrates that a variant is present, it does not address the central question of validating the SENSITIVITY of the primary method of ddPCR that the authors rely on for quantification of the somatic variant. The suggestion of amplicon sequencing was to amplify the region and use NextGen Sequencing (NGS) as a secondary QUANTITATIVE method to validate the sensitivity of the ddPCR results. At the minimum, the authors should clarify how many clones they amplified and sequenced (using SANGER sequencing) to provide an estimate of the sensitivity ddPCR for the same samples. An alternative method would be perform serial dilutions of a plasmid containing the variant with WT plasmids to generate specific populations, 5%, 2%, 1%, 0.5%, 0.1%, 0.05%. Then perform ddPCR on each sample and generate a correlation curve. While the authors cite their own papers in reference to the sensitivity of ddPCR for somatic mutations, in those papers the detection frequencies were far higher (80-90%) and therefore the low sensitivity of the assay was not addressed.

Our response: We appreciate your insightful comments and helpful advice regarding the sensitivity validation of ddPCR for detecting somatic variants. To address this concern, we performed a serial dilution experiment using a plasmid containing the somatic variant mixed with wild-type plasmid to create defined variant allele frequencies (VAFs) ranging from 0.03% to 10%.

The calculated VAFs correlate strongly with the known plasmid VAFs, with an R^2 value of 0.99998, confirming the quantitative robustness of the method across the tested range. The data of ddPCR are summarized below and visualized in the accompanying correlation plot (also included in the revised manuscript):

Real VAF (%)	Fam	Fam+Hex	VAF by ddCPR (%)
0.03	3	10260	0.0292
0.04	4	10740	0.0372
0.05	5	10380	0.0482
0.10	13	12240	0.1062
0.50	57	11890	0.4794
1.00	125	12620	0.9905
2.00	238	11940	1.9933
5.00	596	11860	5.0253
10.00	1026	10280	9.9805

Added in the Methods:**Validation of ddPCR sensitivity**

To validate the sensitivity of ddPCR for detecting low-frequency somatic variants, we generated a series of plasmid mixtures containing the target variant at defined variant allele frequencies (VAFs) of 0.03%, 0.04%, 0.05%, 0.1%, 0.5%, 1%, 2%, 5%, and 10%. These samples were prepared by mixing a plasmid carrying the variant allele with a wild-type plasmid at corresponding ratios. Each sample was analyzed using ddPCR to quantify the observed VAF and assess the assay's sensitivity and accuracy. A threshold of at least three variant-positive droplets was used to define the detection limit.

Added in the Results:

We performed a series ddPCR with low-frequency somatic variants to validate the sensitivity of the system. The ddPCR assay demonstrated robust quantitative performance across a wide range of variant allele frequencies (VAFs) from 0.03% to 10% (Supplementary Figure S2C). Under our defined threshold of at least three positive droplets, the assay reliably detected variants at a minimum VAF of 0.03%. The correlation between the expected VAFs and the ddPCR-measured VAFs was strong, with an R² value of 0.99998, indicating the accuracy of ddPCR in quantifying somatic variants within this range (Supplementary Figure S2C and Supplementary table S8).

Supplementary Figure S2

Figure legends: **A**, *Upper*, plot of ddPCR result on negative control H7; *Lower*, plot of ddPCR result on negative control H14; **B**, Validation of NAP1L1 p.D349E. PCR amplicons were cloned and subjected to Sanger sequencing analysis. The sequencing electrophoretograms are shown (*Left*, wild-type allele; *Right*, mutant allele); **C**, Correlation curve of observed VAFs (y-axis) versus expected VAFs (x-axis) for each plasmid mixture. Plasmid mixtures containing the target somatic variant at defined variant allele frequencies (VAFs) ranging from 0.03% to 10% were analyzed by ddPCR.

Reviewer #2

Comment: First regarding my original comment 6, I was asking about inflammatory cytokines, not simply type I interferons. The IFN ELISA data provided in response to review is convincing, but did the authors see IL-6 or IL-1 protein secretion? As mentioned, cGAS is not a strong inducer of pro-inflammatory cytokines, and IFNa/b are interferons, not inflammatory cytokines. The authors should provide additional clarification and data, as well as including the IFN ELISA data in the revised manuscript, not simply in response to reviewers.

Our response: We appreciate your insightful comment regarding the inflammatory cytokine response. We have now included data for IL-6 and IL-1 protein secretion in the revised manuscript. We have also supplemented the ELISA data of IFN, and the added content is as follows:

Figure legend: Elisa of Ifna, Ifnb, Il-6 and Il-1b in *Nap111* knockdown compared with control group.

Figure legend: Elisa of Ifna, Ifnb, Il-6 and Il-1 β in mice treated with the wild-type and Nap111 p.D349E plasmid.

Figure legend: Elisa of Ifna, Ifnb of the left ventricle tissues from mice treated with wild-type and Nap111 p.D349E AAV.

Figure legend: Elisa of Ifna, Ifnb, Il-6 and Il-1 β of the left ventricle tissues from mice treated with wild-type and Nap111 p.D349E.

Regarding comment 7 and response, I am unclear why the authors chose not to include the data from the NAP1L1 D349E knock-in mouse model, where WT NAP1L1 is ablated. I think these data are supportive of a role for NAP1L1 mutation in hypertrophy and should be included, especially if the cGAS/STING pathway is activated and interferon-related genes are increased. They support the AAV models and do not involve the AngII confounder.

Our response: Thank you for raising the important point regarding the NAP1L1 D349E knock-in mouse model. We agree that data from this model are highly relevant and provide additional support for the role of *NAP1L1* mutations in hypertrophy. After considering your suggestion, we have now included the data from the NAP1L1 p.D349E knock-in mouse model in the manuscript as the Supplementary Figure S7. These data complement our previous findings using the AAV models and offer a valuable perspective, particularly since they do not involve the AngII confounder.

Methods:

Results: In order to avoid the confounding factor of AngII and basal expression of Nap1l1, we generated Nap1l1 p.D349E knock-in (KI) mice (Supplementary Figure S7A and S7B). Nap1l1 p.D349E KI mice exhibited significant enlargement in heart size and cardiomyocyte size (Supplementary Figure S7C and S7D). Nap1l1 p.D349E KI mice also showed more severe myocardial fibrosis (Supplementary Figure S7E and S7F). Nap1l1 p.D349E KI mice exhibited worse cardiac function, including LVPW;d, LVPW;s, EF, FS (Supplementary Figure S7G and S7H). Correspondingly, *Ifna*, *Ifnb* and immune factors Il-6 and Il-1b were increased in Nap1l1 p.D349E KI mice (Supplementary Figure S7I).

Supplementary Figure S7

Figure legends: **A** and **B**, Schematic diagram depicting the experimental strategy used in C57BL6/N mice. **C**, Representative image of heart and WGA staining; **D**, Quantification of WGA staining in the control and KI group. **E**, Representative hematoxylin and eosin staining Masson's trichrome staining; **F**, Quantification showing the fibrotic area in control and KI group. **G**, Representative M-mode echocardiograms of the left ventricle of the control and KI group. **H**, Quantification of left ventricular end-diastolic anterior wall thickness (LVAW; d), left ventricular end-diastolic posterior wall thickness (LVPW; d) ejection fraction (EF), fractional shortening (FS) of the control and KI group; **I**, Elisa of Ifna, Ifnb, Il-6 and Il-1b of the left ventricle tissues of

the control and KI group.

Reviewer #3

Comment: No additional comments to the authors. Care and diligence were taken to address the previous concerns.

Our response: We sincerely thank Reviewer #3 for the positive feedback and appreciate your time and effort in reviewing our manuscript.